# GENERATING EXPLANATIONS FROM LINEAR STRUCTURAL CAUSAL MODELS

## ABSTRACT

Causality and explainability are intertwined in that they mutually inform each other. For instance, incorporating knowledge on the causal structure of the data into an explanation aligns the reasoning within said explanation with how the data was generated. Surprisingly, this conceptual idea of generating explanations mainly from a suitable causal representation, like Pearl's Structural Causal Model (SCM), has not been studied before. To this end we are going to present a first algorithm within this new type of explanation that takes an SCM as input. We start by identifying desiderata for this new approach by discussing shortcomings of previous causal explainers. Our current key restriction are linear SCM, for which we then define the set of possible questions before deriving the actual algorithm step-by-step alongside an example. To better understand whether our so-called Structural Causal Explanations (SCE) are sensible w.r.t. the initial desiderata we asked 22 study participants to provide their guess of causal relations on simple, every-day variables to then evaluate SCE on these SCM approximations. We find that SCE is a suitable explanation scheme and followup our empirical study of SCE with SCM approximations as discovered by popular graph learning algorithms. In this second experiment we find that SCE reveals defficiencies of current graph learning algorithms for which we then propose a naïve regularizer that incorporates SCE into learning.

## 1 INTRODUCTION

Artificial intelligence research dreams of an automation to the scientist's manner (McCarthy, 1998; McCarthy & Hayes, 1981; Steinruecken et al., 2019) and while causal interactions stand at the center of human cognition (Penn & Povinelli, 2007), harnessing its prowess is still an ongoing endeavour (Schölkopf, 2022) even in the face of more recent formalizations of causality (Pearl, 2009; Peters et al., 2017). One key promise of causality for the sake of AI lies in explanations, since causality and explanations are widely regarded as intertwined (Josephson & Josephson, 1996). For example Miller (2019) describes explanations as consisting of two processes and a product. The first process is a cognitive abductive inference step to determine the *causes* for a given event (possibly in reference to a counterfactual). The product is then the actual explanation. The second process is a social part where the knowledge transfer between explainer and explainee occurs (usually people interacting with each other). Now the question arises, what would constitute a suitable representation for such an explanation? In their seminal book, Pearl & Mackenzie (2018) argue that counterfactual, symbolic causal reasoning is the most important factor for machines to achieve true human-level intelligence since it ultimately constitutes the way humans reason and explain. Suggesting that the *structural causal model* (SCM) representation, which is at the center of Pearl's counterfactual theory of causation, is a suitable representation for explanations. Several works in cognitive science are indeed in support of Pearl's formalism as a great tool to capture important aspects of human reasoning (Gerstenberg et al., 2015; 2017) and thereby also how humans provide explanations (Lagnado et al., 2013). Furthermore, questions of the form "What if?" and "Why?" have been shown to be used by children to learn and explore their external environment (Gopnik, 2012; Buchsbaum et al., 2012) and are essential for human survival (Byrne, 2016). These humane forms of causal inferences are part of the human mental model which can be defined as the illustration of one's thought process regarding the understanding of world dynamics (see also discussions in Simon (1961); Nersessian (1992); Chakraborti et al. (2017)). In the following, we will consider the definition of an explanation

to be an answer to a why-question (Dennett, 1989), which is ultimately a counterfactual question (thus a causal notion). Specifically, we will define what particular type of why-questions we answer and devise a new generation scheme for answering such why-questions based on an SCM.

Overall, we make several contributions: (i) we derive a new type of explanation based on SCM (called Structural Causal Explanations, short SCE) by reflecting on conceptual progress in prior literature on combining causality and explanations in AI, (ii) we conduct a human study to assess that SCE is sensible across various examples, (iii) we feed SCE with causal representations learned from data to assess what the explanations reveal about the underlying graph learning methods, and (iv) we present a naive regularization penalty to reduce the number of false links in learned graphs.

We make our code repository publically available at: `https://anonymous.4open.science/r/Structural-Causal-Explanations-D0E7/`

## 1.1 SHORTCOMINGS OF PREVIOUS EXPLAINERS (SHORT CASE STUDY: CXPLAIN)

We start by reflecting on existing key ideas within the realm of explainable AI that makes use of causality. Since a great deal of existing literature is concerned with *causal attribution* we are going to discuss a representative case in the popular approach by Schwab & Karlen (2019) called CXPlain. Fig.1 top shows a why-question in a medical data setting. Particularly, patient Hans' medical condition is captured by different covariates (age, nutrition, health, mobility) and the question is concerned with why Hans' mobility value is lower on average than that of other patients. Fig.1 bottom then shows the answer given by CXPlain assigning three positive numerical scores to all variables except the one in question with the highest value being given to age, then nutrition and finally health. We can interpret this result as saying that all potential factors are actually being deemed relevant and "causal" to Hans' mobility. To briefly explain the setup: Hans' values were sampled from a synthetic SCM which CXPlain had access to while training its surrogate explanation model. We trained 10 bootstrapped neural models using suitable parameters for the masking operation and loss function. Returning to our score distribution, this single observation makes apparent two important shortcomings of such causal attribution explainers: firstly, from the output we cannot deduce which is a direct (health in this case) and which are indirect (age and nutrition mediated via health) causes. Secondly, we have no information on the causal effect, that is, we

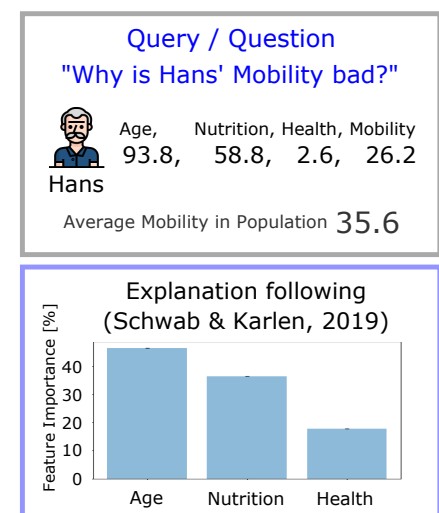

Figure 1: **Conceptual Limitations of Explaining Counterfactual Questions Without Use of SCM.** Refer to the text on the left for a discussion of the limitations. The figure shows a medical record for patient Hans and a question concerning the reasoning behind the sate of Hans' mobility. (Schwab & Karlen, 2019)'s method provides positive feature importance scores for the remaining variables. (Best viewed in color.)

cannot tell in which way a variable with high attribution will affect the predicted variable, for example the nutrition variable received a high importance score than age but age will have a detrimental effect on mobility whereas nutrition will have a beneficial effect. Two further, less important but still noteworthy, shortcomings are the following: thirdly, the attributions are deterministic. This might first be considered a feature, however, the causal mechanism of an SCM are only deterministic up to a realization of the exogenous variables. Therefore, we can have the exact same patient record for *different patients*. This cannot be captured by these previous attribution methods. Fourthly, when querying for random individuals we actually observe inconsistencies between the attributions themselves which is illogical since the patient records are being generated by the same causal mechanisms. For example in the Hans case we had age, nutrition and then health ordered from highest to lowest attribution. For a rather similar patient we observe that nutrition and age swap in importance. For yet another patient we observe that suddenly age and nutrition, which previously played the most important role, are not important anymore. The reader is invited to look at the plots highlighting the last two deficiencies from Fig.6 of the appendix which shows the same setup as from Fig.1 but for

different queries, that is, patients other than Hans. In conclusion, the four discussed deficiencies we pose as *desiderata* for our new approach to resolve. In summary, our approach should ideally: (Desideratum 1) differentiate direct from indirect causes, (D.2) capture qualtitative information on causal effects and (D.3) cope with stochasticity. SCE satisfies the desiderata through its SCM.

## 2 PREREQUISITES & ASSUMPTIONS

We follow the formalism of (Pearl, 2009) for discussing causation but adapt a modern formalization inline with works such as (Bongers et al., 2021). The key input to our explainer is going to be an (approximation of an) SCM, which we define as:

**Definition 1** (SCM). *A structural causal model is a tuple $\mathcal{M} = (\mathbf{V}, \mathbf{U}, \mathbf{F}, \mathrm{P}_{\mathbf{U}})$ forming a directed acyclic graph $\mathcal{G}$ over variables $\mathbf{X} = \{X_1, \ldots, X_K\}$ taking values in $\boldsymbol{\mathcal{X}} = \prod_{k \in \{1 \ldots K\}} \mathcal{X}_k$ subject to a strict partial order $<_{\mathbf{X}}$, where*

- $\mathbf{V} = \{X_1, \ldots, X_N\} \subseteq \mathbf{X}, N \leq K$ *is the set of endogenous variables.*

- $\mathbf{U} = \mathbf{X} \setminus \mathbf{V} = \{X_{N+1}, \ldots, X_K\}$ *is the set of exogenous variables.*

- $\mathbf{F}$ *is the set of deterministic structural equations, $V_i := f_i(\mathbf{X}')$, where $\mathbf{X}' \subseteq \{X_j \in \mathbf{X} \mid X_j <_{\mathbf{X}} V_i\}$ denoted by $\mathrm{pa}(V_i)$ are the parents of $V_i$.*

- $\mathrm{P}_{\mathbf{U}}$ *is the probability distribution over $\mathbf{U}$.*

For the causally curious reader we point to appendix Sec.A.2 for discussions of our framework in the light of unobservable confounders and similar phenomena. To simplify the discussion around causal effects and such later on when deriving SCE, we restrict the class of SCMs to that of the well-studied linear SCMs. Formally, we have:

**Assumption 1** (Linear SCM). *The SCMs under consideration have linear structural equations, that is, $\mathbf{F} \subset \{f \mid f(\mathrm{pa}(v)) = \boldsymbol{\alpha}^\top \mathrm{pa}(v), \boldsymbol{\alpha} \in \mathbb{R}^{|\mathrm{pa}(v)|}\}$.*

While the above assumption will be sufficient for the discussion of discrete random variables, our results naturally extend to continuous random variables w.l.o.g. if the following assumption on the exogenous variables holds:

**Assumption 2** (Gaussian SCM). *The SCMs under consideration have normally distributed exogenous variables, that is, $\mathrm{P}_{\mathbf{U}} = \mathcal{N}(\boldsymbol{\mu}, \boldsymbol{\Sigma})$ with $\boldsymbol{\mu} \in \mathbb{R}^{|\mathbf{U}|}, \boldsymbol{\Sigma} \in \mathbb{R}^{|\mathbf{U}|^2}$.*

Since an explanation demands the discussion of causes, a very useful object for actually capturing the amount of influence a cause has on its effect is the *causal effect* which is defined as the interventional distribution as follows:

**Definition 2** (CE). *For some target of interest $V_i$ and a set of variables $\mathbf{V}' \subset \mathbf{V}$ an intervention $do(\mathbf{V}' = \mathbf{v}')$ replaces all original structural equations $\{f_j\}_{V_j \in \mathbf{V}'}$ by the constant assignment $V'_j := v'_j$. The induced distribution $p(v_i \mid do(\mathbf{v}'))$ is called causal effect of $\mathbf{V}'$ on $V_i$.*

Lastly, to quantify the causal effect for any given random variable pair in a single scalar value, we can resort to the average effect. We have that:

**Definition 3** (ACE). *For a pair of random variables $(V_i, \mathbf{V}') \subset \mathbf{V}$ that satisfy*

- (D) $\forall V_j \in \mathbf{V}', \frac{1}{k-l}(\mathbb{E}[V_i \mid do(V_j = k)] - \mathbb{E}[Y \mid do(V_j = l)]) = \alpha_j$ *with $k, l \in \mathcal{X}_j$ and $\boldsymbol{\alpha} = (\alpha_j)_{V_j \in \mathbf{V}'}$ in the case that $(V_i, \mathbf{V}')$ are discrete random variables or*

- (C) $\frac{\partial}{\partial \mathbf{v}'} \mathbb{E}[V_i \mid do(\mathbf{v}')] = \boldsymbol{\alpha}$ *where $(V_i, \mathbf{V}')$ are continuous*

*we call $\boldsymbol{\alpha}$ the average causal effect of $\mathbf{V}'$ on $V_i$.*

Remember the notation of the expected value for a discrete random variable pair $X, Y$ as $\mathbb{E}[Y \mid x] := \sum_y y \cdot p(y \mid x)$ and for continuous RVs as $\mathbb{E}[Y \mid x] := \sum_y y \cdot p(y \mid x)$, where $\mathbb{E}[Y \mid do(X)]$ then refers to the expected causal effect, that is, the expected value of $Y$ under intervention $do(X)$ replacing the conditional distribution $p(y \mid x)$ with the causal effect $p(y \mid do(x))$. In a last step, we connect our assumptions with the definition of ACE.

**Observation 1.** *In SCMs that follow Assumption 1 the coefficients $\boldsymbol{\alpha}$ of the structural equations are average causal effects since Def.3(D) is satisfied. For continuous random variables SCMs following Assumption 2 is necessary to satisfy Def.3(C).*

This observation tells us that with the above definitions and assumptions when deriving our explanations we will be able to simply treat our SCM as a weighted adjacency matrix of the endogenous variables, which in turn will simplify computation immensely and make our explanations compatible with a wide range of existing graph learning algorithms.

## 3 Deriving a New Type of SCM-based Explanation

While acknowledging the difficulty of the problem and its philosophical nature, we address it pragmatically in a step-by-step derivation that leverages qualitative knowledge on SCMs as defined in the previous section. This connection will justify the naming as *Structural Causal* Explanation. Our running example, that of patient Hans, is a homage to the famous fallacy in explainable AI and psychology known as "Clever Hans" named after the 20th century Orlov Trotter horse Hans that was wrongly believed to be able to perform arithmetic (Pfungst, 1911). A "Clever Hans" moment is failure due to spurious associations in the data. For example, an image classifier that learns on watermarked images will have high accuracy on the test data from the same distribution by predicting the class using the watermark labels (that is, the model is "right for the wrong reasons") and furthermore fails completely when moving out-of-distribution (Lapuschkin et al., 2019). Some works such as (Stammer et al., 2021) moved beyond basic methods (like heat maps for image data) by employing expert intervention to move beyond such "Clever Hans" fallacies. Since explanations ought not only be "clever" but also causal, we will refer to our running example as the "Causal Hans" example. In the following, consider an SCM as before that generates medical records described by numerical representations for age, nutrition, overall health and mobility respectively ($\mathbf{V} = \{A, N, H, M\}$). Next, let's consider some samples from said SCM. E.g. we might observe the data set containing the individual named Hans $\mathbf{H} = (H_A, H_N, H_H, H_M) = (93.8, 58.8, 2.6, 26.2)$ where for sake of simplicity each value could be associated with a discrete label e.g. $H_A = 93.8$ would be 93 years old, whereas $H_M = 26.2$ could refer to a rather immobile person. The latter label is actually implicitly the assessment $H_M < \mu_M$ where $\mu_M = 35.6$ is the population's average mobility value, that is, we observe Hans to be a rather immobile person *when compared (or relative) to the other patients*. The population average values for our running example are $\boldsymbol{\mu} = (\mu_A, \mu_N, \mu_H, \mu_M) = (62.6, 32.8, 45.1, 35.6)$. With this we are in the position to pose a question like

**Q1:** *"Why is Hans's Mobility bad?"*

where the word "bad" refers to "bad relative to the population." Formally, we can now define such a question as:

**Definition 4** (Why Question). *A quantity $Q_i := R(v_i, \mu_i)$ with binary ordering $R \in \{<, >\}$ where $V_i \in \mathbf{V}$ and $\mu_i$ is the empirical mean value for $V_i$, is called why-question concerning $V_i$ if the ordering holds true, that is, $\mathbb{1}_R(Q_i) = 1$.*

Remember the notation for the indicator function $\mathbb{1}_R(Q_i) = 1$ if $(v_i, \mu_i) \in R$ and 0 otherwise. Checking back with the definition, we see that **Q1** defines a valid question for the Causal Hans example since $Q_M := H_M < \mu_M = 26.2 < 35.6$ holds true in our example data. On another note, we call $Q_i$ a why-question because it relates to the *counterfactual* scenario regarding the causes of $V_i$, for example, how would've age, nutrition and health had to be if we were to think that Hans' mobility was not bad. While the number of valid why-questions that can be asked seems limited at first sight, the number scales linearly with the SCM as it is coupled to the endogenous variables. Specifically it is $\mathcal{O}(|\mathbf{V}|)$ thus we can potentially ask a question for any endogenous variable of an arbitrarily large SCM.

Next, we will discuss the knowledge on the SCM that our explanation will leverage. Generally, the true data-generating SCM $\mathcal{M}^*$ is unobserved but we can realistically expect to have access to an estimate of $\mathcal{M}^*$. Let's consider following SCM estimate $\mathcal{M}$ that contains the relations $A \xrightarrow{\alpha} N$, $A \xrightarrow{\beta} H$, $N \xrightarrow{\gamma} H$, $H \xrightarrow{\delta} M$ where $\alpha, \beta, \gamma, \delta$ denote the respective (average) causal effects. Further, $\alpha, \gamma, \delta > 0$ while $\beta < 0$. That is, $\alpha > 0$ means that increasing age by a single unit increases nutrition by $\alpha$ units. Similarly, $\beta < 0$ means that for any unit increment of age we will have a $\beta$

number of units decrement of health. Furthermore comparing between coefficients, $\beta > \gamma$ means that the causal effect of aging on health is greater in absolute terms than the causal effect of nutrition onto health. Now when we intend on answering **Q1** it seems reasonable to start with the queried variable first, mobility in this case. Since we know that $M$ is an effect of $H$ with $\gamma > 0$ we expect the below average mobility to be explained by an already below average health value. Indeed, this expectation is met since $H_H < \mu_H$. Traversing the chain further to the causes of $H$, which are $A, N$, we observe two different scenarios. Since $A$ is above average as Hans is an elderly person ($H_A > \mu_A$) and $\beta < 0$ we can conclude that $H_A$ is definitely an explanation for $H_H$ whereas $N$ with $\gamma > 0$ is actually a countering factor since Hans has a good diet ($H_N > \mu_N$) beneficial to his health. In summary, by exploiting the knowledge on $\mathcal{M}$ we have arrived at a causal explanation that can be pronounced in natural language as:

**Explanation 1** (for **Q1**). *"Hans's Mobility is bad because of his bad Health which is mostly due to his high Age although his Food Habits are good."*

Explanation 1 is indeed an explanation as required by the definition of (Dennett, 1989) since it is an answer to the why-question concerning Hans' mobilitity. Furthermore, it is a causal explanation since the used coefficients for deriving the explanation are based on SCM $\mathcal{M}$ that satisfies the assumptions from Sec.2 thus qualifying the coefficients as causal effects. Our above explanation captures two prominent modes of human reasoning, namely both the existence and the "strength" of a causal relation. In the following we will capture and formalize our intuition that allowed us to derive Exp.1, which in turn allows us to compute such causal explanations automatically.

When reflecting on the actual knowledge used in our argument above, then we realize that we can abstract away four key aspects: (i) that there is a relative notion in the why-question $Q_M$ like "why …bad?" that implicitly compares an individual (here, Hans) to the remaining population of patients, (ii) the causal graph provides the structure of the explanation by following any previously unexplained directed path to the target effect (here, mobility), (iii) the causal effect for any pair allows us to assert whether the observed values for that pair are "surprising" in that they are consistent with the mechanisms of the data-generating process or not, and (iv) that some causal effects are more important or influental than others (here, age versus nutrition w.r.t. health). Following this reflection we define sth. called *causal scenario* that will cover (i-iii) as point (iv) will be covered separately.

**Definition 5.** *As before let $V_i, V_j \in \mathbf{V}$ and $\alpha$ denote the ACE from $V_j$ onto $V_i$ and $\mu_i, \mu_j$ are the averages of our data sample. The tuple $\mathbf{C}_{i,j}:=(\alpha, v_i, v_j, \mu_i, \mu_j)$ is called a causal scenario.*

With this convenient notation at hand, we are ready to abstract the logic of our explanation to general rules. For this we will make use of first-order logic.

**Definition 6** (Explanation Rules). *Let $\mathbf{C}_{i,j}$ denote a causal scenario, $R_i \in \{<, >\}$ be a binary ordering relation and $\boldsymbol{\alpha}_i^{\mathrm{pa}}$ be the set of all absolute parental ACEs onto $V_i$. We define FOL-based rule functions as followed by indicating for each rule ERx: $(\mathbf{C}_{i,j}, R_1, R_2, \boldsymbol{\alpha}_i^{\mathrm{pa}}) \mapsto \{-1, 0, 1\}$ how the causal relation $V_i \leftarrow V_j$ satisfies that rule.*

*(ER1)* If $R_1 \neq R_2$, then: $((\alpha < 0) \wedge \delta_1) \vee ((\alpha > 0) \wedge \delta_2)$

*(ER2)* If $R_1 \neq R_2$, then: $((\alpha > 0) \wedge \delta_1) \vee ((\alpha < 0) \wedge \delta_2)$

$$\text{with } \delta_1 := R_2(v_j, \mu_j) \wedge R_1(v_i, \mu_i) \text{ and } \delta_2 := R_2(v_j, \mu_j) \wedge R_2(v_i, \mu_i)$$

*And as an extra, "modifier" rule in the case where $|\boldsymbol{\alpha}_i^{\mathrm{pa}}| > 1$ we simply consider the parent with the highest ACE absolutely, $V_k^* = \arg\max_{V_k \in \mathrm{pa}(V_i)} \boldsymbol{\alpha}_i^k$, as the most important direct cause.*

Since our rules only need qualitative knowledge on the causal effects (i.e., we simply test the sign of the coefficients) it is possible to use techniques from *partial identification* which allows for bounding causal effects using fewer necessary assumptions at the price of exact estimation (Balke & Pearl, 1994). These two plus one rules build the foundation for our new type of explanation. Having the actual relation $R$ as a return argument of each of the rules allows for a fine-grained explanation. In a nutshell, it allows to extend a statement "$V_i$ because of $V_j$" to a more detailed one like "$V_i$ because of $V_j$ *being low*". The general pronunciation scheme for the the three rules which we name excitation (*ER1*), inhibition (*ER2*), and preference (*ER3*) are summarized in Tab.1. The pronunciation of the details to the relation e.g. "low"/"high" is context-dependent in that these words might need to

| | | |
|---|---|---|
| *ER*1 | Excitation | "$V_i$ because of $V_j$ [being low/high]" |
| *ER*2 | Inhibition | "$V_i$ although $V_j$ [is low/high]" |
| *ER*3 | Preference | "mostly" + *ER*1 or *ER*2 pronunciation |

Table 1: **Pronunciation Scheme.** Right shows the natural language reading of a rule's activation.

replaced with adequate/corresponding words suitable for the context. To elaborate, "the mountain top is cold because of the high altitude" is fine, while "the remaining car fuel is low because of the driver's bad driving style" requires the context-adaptation (what was "low" previously is "bad" in this case). Another noteworthy detail to the SCE properties is the property of *non-repeating causes within explanations* which reduces redundancy. Consider for instance our lead example on Hans's mobility (Exp.1), the SCM suggests that $N$ can also be explained by $A$, since $A \rightarrow N$. However, the corresponding SCE does not give this reason because of the aforementioned property which ensures that redundancy is being avoided. I.e., in the explanation step before we actually explain $H$ using both $A$ and $N$, since $\{A, N\} \rightarrow H$, therefore, making it irrelevant for the question to explain the relation between the parents $(A, N)$. While we provided intuition on the derivation of these basic FOL rules alongside the "Causal Hans" example, we now additionally motivate the namings "excitation", "inhibition" and "preference". We took inspiration from *neuroscience*, where the former two terms relate to the way neurons interface with each other using their synaptic-dendric connections (He & Cline, 2019). The last term is a term to propose "relativity" and thus a preference for one cause over the other. Returning to our derivation, to now show how these rules can generate something like Exp.1, we will present our actual algorithm definition. We define the Structural Causal Explanation algorithm as:

**Definition 7** (SCE). *Like before let $Q_i, \mathcal{M}$ be a valid why-question and some SCM estimate respectively. Further, let $\mathbf{D} \in \mathbb{R}^{n \times |\mathbf{V}|}$ denote our $n$-samples data set. We define a recursion*

$$\mathbf{E}(Q_i, \mathcal{M}, \mathbf{D}) = (\bigoplus_{V_k \in \text{pa}(V_i)} ER(V_i \leftarrow V_k), \bigoplus_{V_k \in \text{pa}(V_i)} \mathbf{E}(Q_k, \mathcal{M}, \mathbf{D})) \quad (1)$$

*where $ER(V_i \leftarrow V_k)$ is a shorthand for $\{ERx(\mathbf{C}_{i,j}, R_1, R_2, \boldsymbol{\alpha}_i^{\text{pa}})\}_{i=1,2,3}$ that are given through $(\mathcal{M}, \mathbf{D})$ and $\bigoplus_{i=1}^{n} v_i = (v_1, \ldots, v_n)$ denotes concatenation. The recursion's base case is being evaluated at the roots of the causal path to $V_i$, that is, for some $V_k \in \mathbf{V}$ with a path $V_k \rightarrow \cdots \rightarrow V_i$ we have*

$$\mathbf{E}(Q_k, \mathcal{M}, \mathbf{D}) = \emptyset. \quad (2)$$

*We call $\mathbf{E}(Q_i, \mathcal{M}, \mathbf{D})$ Structural Causal Explanation for $V_i$ based on $(\mathcal{M}, \mathbf{D})$.*

The above algorithm is a simple recursion that traverses all possible directed, causal paths to the target variable checking each of the rules *ERx* thus constructing a unique code that maps to a unique answer following our previous pronunciation scheme. Since SCE answers a why-question, which is counterfactual by nature, and does so by using qualitative knowledge of the SCM, which encodes counterfactual knowledge, we can generally classify SCE as a *counterfactual*-type of explanation in the broader scope of conceptually distinct ideas in causal explainable AI.

**Reconstructing the Causal Hans Example using Def.7:** To return one last time to our running example, we apply the recursion step-by-step now. For $Q_M$ (corresponding to **Q1**) we arrive at:

$\mathbf{E}(Q_M, \mathcal{M}, \mathbf{D})$
$= ((ER1 = -1), \bigoplus_{V_k \in \{A, F\}} \mathbf{E}(Q_H, \mathcal{M}, \mathbf{D}))$
$= (\ldots, (((ER1 = 1, ER3 = 1), \mathbf{E}(Q_A, \mathcal{M}, \mathbf{D})),$
$\qquad ((ER2 = 1), \mathbf{E}(Q_F, \mathcal{M}, \mathbf{D})))),$
$= (\ldots, ((\ldots, \emptyset), (\ldots, \emptyset))).$

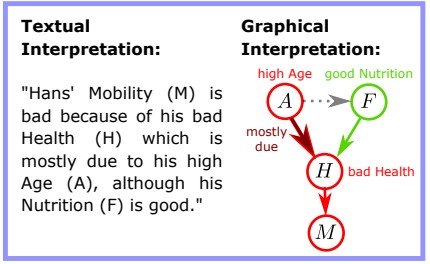

Figure 2: **Interpreting the SCE Output for Causal Hans.** The example illustrates all three rules. (Best viewed in color.)

So the recursion result is $H \rightarrow M : (ER1 = -1, ER2 = 0, ER3 = 0), A \rightarrow H : (ER1 = 1, ER2 = 0, ER3 = 1), F \rightarrow H : (ER1 = 0, ER2 = 1, ER3 = 0)$. This result *uniquely* identifies the human-readable pronunciation of our causal explanation in Exp.1. For the graphical interpretation refer to Fig.2 which highlights the recursive traversal through each pair of parents while avoiding redundancy through duplicate paths.

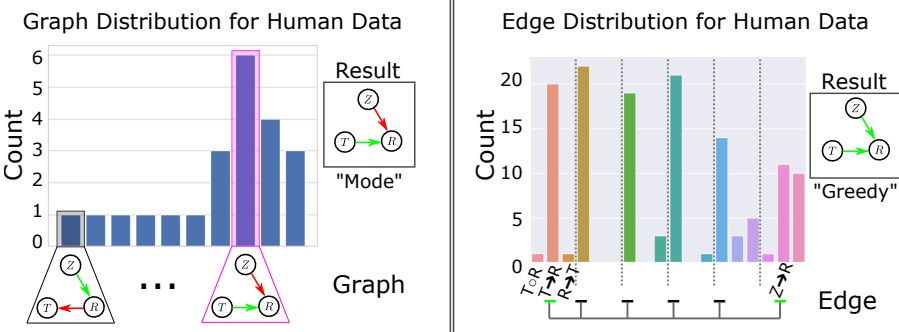

Figure 3: **Measuring Agreement Between Beliefs of Human Subjects Regarding Causal Graphs.** Left, the graph estimate is the mode of the distribution of all predicted causal graphs. Right, greedily pick each edge of the graph. (Best viewed in color.)

## 4 EMPIRICAL STUDY

**Are SCEs Sensible Explanations?** To get an understanding of whether SCE are sensible beyond the running example we have shown, we have conducted a user study with $N = 22$ human subjects that had to judge the qualitative causal structure of four "daily-life" examples using a questionnaire specifically designed to provide us with the data necessary for constructing causal graphs representative of what the participants think about the presented concepts. Please refer to the Appendix for the questionnaire and [Human Data] for the anonymized answers that we used for evaluating the survey. The first question to answer is: how did we construct the graph estimates from human data? In Fig.4 we show two ways that we considered: the "Mode" refers to the scheme where we simply look at the different graphs and take the *most frequently reoccurring* graph as representative of the population, or the "Greedy" approach where we look at the frequency at which edges are predicted and then simply construct a graph from greedily taking the *most probable edge each time*. Greedy comes at the cost that the predicted graph is not necessarily within the populaton. With the human causal graphs at hand, we now investigate our intial question about SCE. For brevity, we will only highlight the most important key observations with a prolonged discussion being provided in the Appendix: Observation **(i)** the SCEs that we generate from the acquired causal graphs are sensible in the sense that they lie close (or are even identical) to the apriori expectation of the study (the proposed ground truth). Observation **(ii)** we observe a systematic approach and thereby non-random approach to edge-/structure-selection by the subjects. Furthermore, there are only a few clusters even with increasing hypothesis space. Both the systematic manner and the tendency to common ground are evidence in support prior evidence that SCMs are a suitable representation for human causal modelling. Observation **(iii)** we observe that the increase in hypothesis/search space (i.e., more variables) comes with an increase in variance. This variance increase can be argued to be due to the progressive difficulty of inference problems as well as decreased levels of attention and potential fatigue across the duration of the experiment, and observation **(iv)** some subjects implicitly assume a notion of time by assuming a cyclic relationship between e.g. treatment and recovery, where the subject likely thought in terms of 'increasing treatment increases the speed of recovery *which subsequently* feeds back into a decrease of treatment'.

**What Can SCEs Reveal About Graph Learning Methods?** Having established the SCE algorithm as a sensible way for producing explanations, the natural next step is to conisder how we can incorporate SCE into learning. To this end, we start by considering SCEs based on popular graph learning methods. Induction of inter-variable relationships based on available data, especially of directed acyclic graphs (DAGs), is paramount in causality (Pearl, 2009). Unfortunately, due to the combinatoric nature of the problem setting, learning DAGs from data is recognized to be an NP-hard problem (Chickering et al., 2004). However, several works have tackled this difficult problem and one solution for learning linear DAGs came from a method called NOTEARS (abbreviated NT, Zheng et al. (2018)) who were able to re-formulate the traditional view into a continuous shape such that any non-convex optimization can be applied for the graph estimation problem. The authors propose the general formulation, $\min_{\mathbf{W} \in \mathbb{R}^{d \times d}} \quad f(\mathbf{W}) \quad \text{subject to} \quad h(\mathbf{W}) = 0$, where $f$ is a data-based score, e.g. a regularized least-squares loss is applied assuming a sparse linear model

(possibly SCM). That is $f(\mathbf{W}) = ||\mathbf{X} - \mathbf{X}\mathbf{W}||_F^2 + ||\mathbf{W}||_1$, and $h$ is a smooth function with a kernel (or null space) that only contains acyclic graphs, $h(\mathbf{W}) = 0 \iff \mathbf{W}$ is acyclic. Different variations of the same continuous counting mechanism using this acyclicity constraint have been proposed, e.g., Zheng et al. (2020) proposed $h(\mathbf{W}) = \text{tr}(e^{\mathbf{W} \circ \mathbf{W}}) - d$ while Yu et al. (2019) proposed $h(\mathbf{W}) = \text{tr}[(\mathbf{I} + \mathbf{W} \circ \mathbf{W})^m] - m$. Unfortunately, both suffer from cubic runtime-scalability in the number of graph nodes, $O(d^3)$. While the aforementioned works have focussed on data originating from (non-linear transformation) of linear SCM, there exists yet another sub-class of DAG-learning methodologies that focuses on more general causal inference. Ke et al. (2019) made use of interventional data to update their graph estimate while using masked neural networks to mimic the structural equations. Brouillard et al. (2020) follows the same idea of leveraging causal information, e.g. interventional data, for overcoming identifiability issues while staying close to the continuous optimization formalism introduced by NT. Returning to our question: we looked at different data sets including different graph learners and for each combination generated their respective SCE. We considered several different why-questions for each of the four data sets: data set for the Causal Hans example, weather forecast (W, real world, Mooij et al. (2016)), mileage (M, synthetic), and recovery (R, real world, Charig et al. (1986)). To avoid cluttering in the main text we have moved the relevant tables and figures to the the Appendix where we also provide an extended account, here we highlight the most important insights (based on graphs from NT): Observation **(i)** matched expectations on the W and M data sets, whereas differences on the R and H data sets. For R, the difference is only subtle as the model's explanation to the why-question "Why did Kurt not Recover?" is not "Kurt did not Recover because of his bad Pre-condition, although he got Treatment." but "[...], which were bad although he got Treatment." which is on the second recursion in the reasoning process i.e., the treatment countering the state of condition and not affecting the condition itself. This difference becomes apparent in the graphical structure where the arrow from Pre-conditions to Treatment is inverted contrary to expectation. To illustrate one more drastic example using the data set of our Causal Hans example, here the discrepancy revolves around a totally different graph structure e.g. the learned model expects a direct cause-effect relation between age and mobility while also wrongly assuming that food habits have a detrimental effect on health. Therefore the answer to the question "Why is Hans's Mobility bad?" suddenly becomes "Hans's Mobility, in spite his high Age, is bad mostly because of his bad Health which is bad mostly due to his good Food Habits." which sounds very absurd. The ground truth SCM for this data set contains non-linear causal relationships, while NT makes linearity assumptions, which explains the wrongly learned graph structure. Observation **(ii)** only by looking at the SCE, effectively using it as a graph distance or metric, we were able to tell that the learned model is very different from what we had initially expected. Put differently, it made apparent for the Causal Hans example that by simply adding an extra edge (here $A, M$) and flipping another (here $N, H$) we already get a big difference in what these graphs express/explain.

**Does a SCE Regularization Penality Improve Graph Learning?** While we have seen that SCEs are sensible explanations but that graph learning methods are still far from perfect in predicting graphs from data, in this final experiment we investigate how to use SCEs *to improve* learning. Since SCEs contain (some) knowledge on causal relationships underlying the data, they should help in improving the overall prediction and sample efficiency of graph learners. We take NT again as graph learner and add a simple regularization term to its loss that penalizes inconsistent explanations. We generate 70 random linear SCMs with respective observational distributions. Then we use graph learning to infer 70 more graphs, making 140 graphs in total. For each graph we generate 50 random why-questions to be answered, resulting in a data set of 7,000 explanations. All the details regarding this learning setup, such as for instance how to make make SCE differentiable for it to function as training signal, are being discussed in the Appendix. The graph learning is being performed in a data scarce setting with only 10 data samples per graph. Thus to infer the true causal structure the method ideally needs to perform sample-efficient learning. Fig.4 shows our results. The error distributions over all of the graphs are shown both with and without the SCE regularization. We also highlight the graph estimate upon which most improvement was observed. It can be observed that with the regularization the method can both identify more key structures while significantly reducing the number of false positives. For example many false links that pointed towards node H (like B to H or G to H) were removed while some key structures could now be recovered like the directed edge from node I to node A. While more experiments would be necessary to claim that indeed learning is (significantly) improved through explanations, our naïve learner already provides evidence in favor of our initial hypothesis.

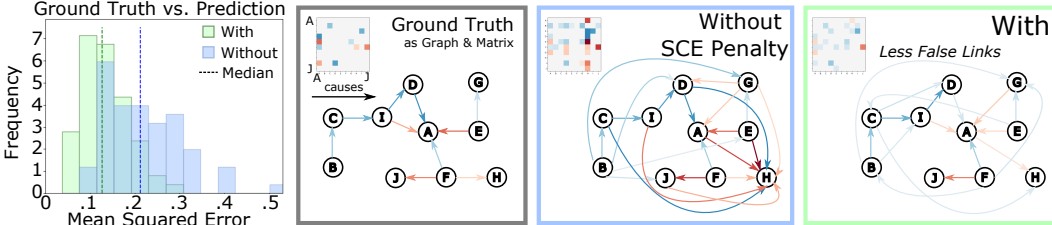

Figure 4: **Graph Learning Improves with Explanations.** Left: error distributions when performing graph learning with/-out SCE regularization (which is simply an added penalty term for inconsistent explanations), next to is the ground truth graph. Right (boxes): the predicted graphs, showing a decreased number of false positives. (Best viewed in color.)

## 5 RELATED WORK

A great body of work within deep learning has provided visual means for explanations of how a neural model came up with its decision i.e., importance estimates for a model's prediction are being mapped back to the original input space e.g. raw pixels in the arguably standard use-case of computer vision (Sundararajan et al.; Selvaraju et al., 2017; Schulz et al., 2020). To circumvent explanations that are like "children that are only able to point fingers", Stammer et al. (2021) proposed a neuro-symbolic explanation scheme to revise behavior from learned models in an interactive loop following the framework of (Teso & Kersting, 2019). On the causal end, (Schwab & Karlen, 2019) proposed a model-agnostic approach that can generate explanations following the idea of Granger causality (which is very different from Pearlian causality as it captures "temporal relatedness" which holds in their setting as input precedes output). On the Pearlian side of explanations, the computation of Causal Shapely Values (Heskes et al., 2020) or the LEWIS framework (Galhotra et al., 2021) are explainers for numerical attribution that capture important distinctions within causality such as direct vs. indirect causes or the necessity-sufficiency distinction of causes. Closest to our work on a semantic level within Pearl's causal framework are arguably works on fairness (Kusner et al., 2017; Plecko & Bareinboim, 2022). For instance, Karimi et al. (2020) investigated how to best find a counterfactual that flips a decision of interest e.g. an applicant for a credit is rejected and the question is now which counterfactual setting (changes to the applicant) would have resulted in a credit approval. Considering unit-level instead of population-level causality, our work can compare to the definitions of Halpern (2016) for "actual causation", where the key difference lies in the *relativity* of our explanation approach to a given sample population in addition to the overall less philosophical approach to causal explanations that shows in both how we generate the explanations and then use them for learning.

## 6 CONCLUSIONS & FUTURE WORK

We've presented a conceptually new approach to causal explanations based on SCMs. By first discussing desiderata that followed from shortcomings of previous explainers from the literature, we then derived an algorithm from first principles using our Causal Hans example. In our empirical section we investigated the quality of SCEs and their integration with learning, which proved successful. Finally, we reflected on the placement of SCE within the broader literature.

Since this work poses an arguably original approach to explanations, there is naturally a lot of opportunity for future work. A first natural step is the extension of SCE to time-series data, which would require a generalization of our definitions of a why-questions, the explanation rules and the actual SCE algorithm. An immediate difficult for such an extension would be the recursive explosion w.r.t. number of time steps measured, that is, how to handle redundancy and the scope of an explanation. Another route for future research would be the relaxation of the linearity assumption, extending SCEs to non-linear SCM. On a conceptual note, making quantitative use of the knowledge on causal effects instead of purely qualitative knowledge would likely allow for more expressive explanations. Finally, we believe that interactive approaches to explanations are a promising paradigm for the future and integrating SCE with XIL (Pfeuffer et al., 2023) seems valuable.

REPRODUCIBILITY

We acknowledge the significance of reproducibility in scientific research and have taken multiple steps to ensure the strength and replicability of our work.

**Code:** Our implementation is accessible on GitHub at `https://anonymous.4open.science/r/Structural-Causal-Explanations-D0E7/`. We have used publicly available software and libraries to guarantee accessibility and have comprehensively described the architecture, software, versions, and hyperparameters in the Appendix. Our code is deterministic, incorporating seeds for all random number generators to guarantee the replicability of results. We attempted to include most of the code used to create the result tables and figures in this manuscript.

**Datasets:** This study only utilizes publicly available datasets which have been correctly cited. Furthermore, the authors contribute to an open source repository containing all the datasets used in this work, which will be made available upon acceptance.

**Algorithm Details:** We have provided thorough descriptions and formulations of our algorithm in the main text, supplemented by additional clarifications, and implementation details in the Appendix, ensuring a clear understanding of our contributions and facilitating reproduction. This documentation is intended to provide researchers with all the necessary information for an accurate replication of our experiments.

**Limitations.** We've highlighted our key assumption of linearity of structural equations, which is an arguably common assumption in causality, in our second main paper section. The second assumption of Gaussianity is only necessary when further considering continuous instead of discrete random variables.

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

# A   Appendix for ICLR 2024 Submission "Generating Explanations From Linear Structural Causal Models"

**Contents**

## A.1   A Primer on SCMs and Mental Models

As a question of cognitive science and psychology, we have placed this section in the appendix for the interested reader.

**Our Hypothesis that SCMs are a Suitable Representation of Mental Models.** It has been argued that at the core of a human mental model (abbreviated MM in the following) the illustration of one's thought process (regarding the understanding of world dynamics) is to be found (Simon, 1961; Nersessian, 1992; Chakraborti et al., 2017). The difficulty of said thought process illustration is partly due to circular and abstract terms like explanation and interpretations for which we do not provide an explicit definition as this is up to philosophical debates and ideally we keep the idea more general than what has been done previously in explainable AI/ML where "explanation equals pixel attributions" in many cases. Assuming the world dynamics to be governed by causality we observe that humans are capable of modelling both causal relationships between endogenous variables and additionally information on the strength of said relationship. Put differently, MM model a causal graph and corresponding causal effects akin to the formal notions from the previous section. Consider the following real world example:

> **MM Example.** *At any given time a human has a state of overall health (relating to fat-muscle ratio, allergies and diseases, etc.) and mobility (relating to the general freedom and flexibility of movement, e.g., a gymnast is more mobile than the average person). Now, the MM allows inferring (1) that mobility is being (partially) caused by something else (for instance health, e.g., being overweight decreases one's mobility) and (2) that different events can have different "strength" e.g., that an average car accident causes more harm to the individual's mobility than an average workout session causes good.*

A natural candidate for capturing the two properties from the MM example formally are SCM, thereby we hypothesize the following:

**Hypothesis 1** (**MM Conversion, short MMC**)**.** *The parts of the MM that are being used for encoding the causal relationships of the variables of interest can be formally captured by a corresponding SCM, in short this "equivalence" can be denoted as MM $\equiv$ SCM.*

While the MMC hypothesis leaves room for notions not captured by mathematical rigor, it suggests an equivalence to SCM regarding the causal aspects. The MM example has motivated the MMC hypothesis which itself suggests *a justification of using SCM in the first place*.

**Implications of MM $\equiv$ SCM**. If we accept that MM $\equiv$ SCM, then we can use SCMs as an adequate proxy to the MM. Furthermore, any useful property of SCM implies corresponding aspects back in the MM. We immediately observe one such key property of SCM namely comparability. That is, if one is given say two different SCMs that are defined over the same endogenous and exogenous variables (so only differing in the actual parameterizations) then one can compare said SCMs i.e., there exists a notion of distance. For the linear case, we can easily prove this by constructing an example metric space.

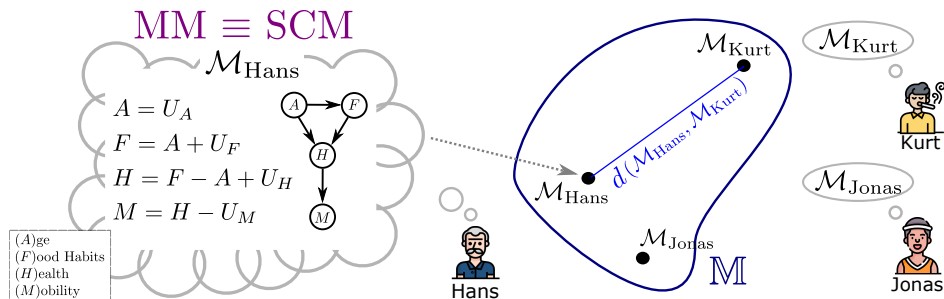

Figure 5: **MMC Hypothesis and Linear SCM Metric Space.** Left: Accepting Hyp.1 means that the MM of Hans is an SCM. Right: Different linear SCM (from different individuals) can be compared, an example metric space for $(\mathbb{M}, d)$ is given by Prop.1. (Best viewed in color.)

**Definition 8.** *We define a function $d(\mathcal{M}_1, \mathcal{M}_2) = \sum_{i \neq j} |\mathcal{M}_1(j,i) - \mathcal{M}_2(j,i)| + q(P_1, P_2)$ where $q$ is the square-root of the Jensen-Shannon Divergence (JSD), $\mathcal{M}_k = \langle \mathbf{U}_k, \mathbf{V}_k, \mathcal{F}_k, P_k(\mathbf{U}_i) \rangle$ for $k \in \{1,2\}$ such that $\mathbf{V}_1 = \mathbf{V}_2$, $\mathbf{U}_1 = \mathbf{U}_2$, $\mathcal{F}_k$ define linear functions in $\mathbb{R}$, and in slight abuse of notation $\mathcal{M}_k(j,i)$ is the causal effect $\alpha$ from $V_j$ to $V_i$.*

**Proposition 1.** *Let $d$ be as in Def.8 and let $\mathbb{M}$ denote the set of all linear SCM defined over the same exogenous and endogenous variables, $\mathbf{U}, \mathbf{V}$. Then $(\mathbb{M}, d)$ is a metric space.*

*Proof.* The absolute difference on the real numbers is a metric (i.e., positive-definiteness, symmetry, and triangle-inequality hold) therefore holding for the "dependency" terms from $\mathcal{F}$. Furthermore, $q$ was chosen as the Jensen-Shannon-Metric. Finally, metrics are closed under summation. □

Prop.1 is just one example of what might be considered a sensible metric space for a subset of all SCMs. What it does is compare each of the linear coefficients for any causally related tuple of variables, aggregating the sum, and further adding a divergence term between the defined distributions over the exogenous variables. This comparability and the visual intuition behind MMC are illustrated in Fig.5. We now state our **first key observation** following Hyp.1 and Prop.1: *the existence of a "true" SCM is in fact justified i.e., there exists an underlying data generating process for any data and the MM of any person might or might not coincide with that SCM.*

On another note, consider the fact that while the "true" SCM represents the concept of objectiveness, oppositely, the MMs are of subjective nature (that is, every human has their own subjective life experience). Coming back to MM ≡ SCM, we see that Prop.1 further implies that MMs are also capable of dis-/agreeing with each other. With this at hand, we now state our *second key observation*: in most practical cases having access to many SCM-encodings of subjective MMs can ultimately lead in their overlap-agreement to (parts of) the objective "true" SCM. There is certainly no guarantee since all available MM-SCM samples can in fact be wrong, however, note the emphasis on *in most practical cases*—therefore, identifying this overlap in MM (or SCM) for a specific problem is highly valuable for AI/ML research.

Our final, *third key observation* is concerned with explanations. Existing literature views explanations as *derivable* from MMs and thus implicitly containing some information on the MM (Chakraborti et al., 2017) and since MM ≡ SCM, we argue that there must exist an equivalent of the human notion of explanation *within SCM*. This justifies our further investigation on SCM-based explanations, which eventually leads to the formalism of SCE. The benefits of an approach using explanations derived from SCM are two-fold (1) that by construction they are human understandable allowing for explainable ML in which models can reason about the learnt and (2) that the models themselves become better, as they need to account for consistency in explanations, which is beneficial to any downstream-task.

## A.2 MULTIPLE NOTEWORTHY SHORT DISCUSSIONS

**The Importance of MM ≡ SCM for SCE.** While the MMC is a fundamental question that cuts to the core of human thinking and remains to be proven right or wrong (although we believe it to

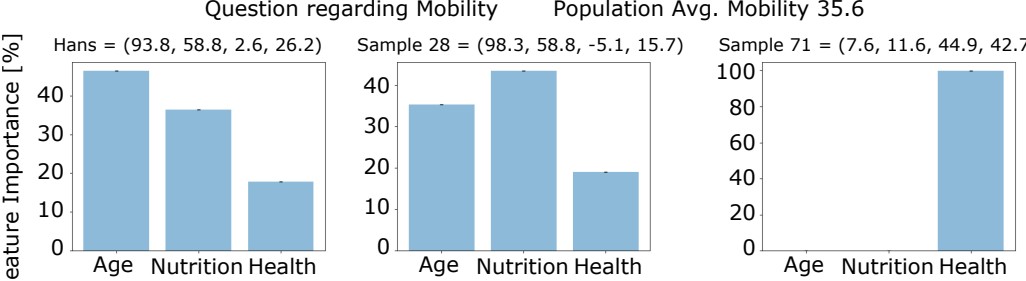

Figure 6: **Further Examples of CXPlain Shortcomings.** Refer to the main text for details.

be true to the extent of representability through SCM), and while we used it to ultimately justify the usage of SCM to then derive the causal explanations we call SCE, still, to the actual existence and formalism of SCE the MMC's truth value is invariant. Put blantly, if the MMC were to be wrong, then the formalism of SCE and all proven properties remain *the same*. However, if MMC were to be true, then SCE in fact become a "stronger" formalism for causal explanations since they'd have a direct link to the MM. More importantly, one could make the case that they'd represent a "natural" formal pendant to the vague human explanations.

**Simpson's Paradox Example.** Consider the well-known Simpson's paradox example for the medical setting of Kidney stone treatments from (Charig et al., 1986). The setting is given by $T, K, R$ which are Treatment, Kidney Stone Size, and Recovery respectively, and further the graph is given by $T \rightarrow R, K \rightarrow \{T, R\}$. It is known that $T = 0$ denotes open surgery and $T = 1$ denotes Percutaneous nephrolithotomy (being a more involved procedure) and in the overall statistics for recovery of the patient (denoted by $R = 1$) we observe $78\%$ versus $83\%$ respectively, suggesting that $T = 1$ is the better option. Yet, when looking at the confounder $K$ values of patient recovery, we observe $93\%$ versus $87\%$ for a small kidney stone $K = 0$ and $73\%$ versus $69\%$ for a large kidney stone $K = 1$ respectively, suggesting that in fact $T = 0$ is better instead. This is the "paradoxical" situation, which is sensible from the *causal perspective*. If we now ask the single why-question for patient $i$ with say values $T = 1, R = 0, K = 1$ on why $i$ did not recover $r_i < \mu^R$ (where $\mu^R$ is the mean recovery of the data set), then we obtain an SCE that reads as follows: *"Patient $i$ did not recover because of the large kidney stone, although (s)he had Percutaneous nephrolithotomy."*

**Hidden Confounders in Semi-Markovian Models.** As we pointed out in the main text, SCE can naturally handle/extend to semi-Markovian models. For illustration, consider the non-Markovian alternative to the example from the paragraph above on Simpson's paradox, where $K$ is a hidden confounder i.e., we only observe $T \rightarrow R$ as the graph. In a lot of practical settings we might at least be aware of the fact *that there is* hidden confounding present between the two variables and thus have an additional (dashed) bi-directed edge between $T$ and $R$ (case 1) and in the arguably worst case, said variable is fully undetected (case 2, in this case it is not necessarily a hidden confounder but simply a hidden cause, since we don't know if it is confounding or not—confounding meaning the same thing as *common cause*). Let's consider both cases, in case 1, the SCE for the same question as before would read as: *"Patient $i$ did not recover although (s)he had Percutaneous nephrolithotomy."* We note that simply the reasoning on $K$ is not being delivered, naturally, since $K$ is not in the SCM that the SCE process observes. For case 2, we'd observe the same reading due to the definition of the SCE construction. Here, however, we note that this case allows for a natural extension of SCE in which the reading could change to possibly, "Patient $i$ did not recover because of *an unknown reason*, although (s)he had Percutaneous nephrolithotomy." Note that this semi-Markovian SCE now allows for reasoning with "unknown reasons" since the hidden cause $K$ will certainly have a causal relation to $R$ (since $K$ is a cause) but the name of $K$ will not be revealed (since $K$ is hidden). With this example, we thus conclude that Markovianity can be leveraged by SCE.

**Algorithm for SCE Regularization.** For sake of completion, we provide an explicit example algorithm for the simple penalty term we added in our setup for improving graph learning with available SCEs for the respective data set. For particular details consider the following section on "Details for Using SCE as Regularizer". When we write "SCM $H$" then what is meant is the

weighted adjacency matrix where the weights represent the causal effects (that is, coefficients in the linear structural equations) due to Assumption 1. Therefore, the causal graph can simply be extracted through $|\tanh(H)|$.

**Technical Details and Code.** All experiments are being performed on a MacBook Pro (13-inch, 2020, Four Thunderbolt 3 ports) laptop running a 2,3 GHz Quad-Core Intel Core i7 CPU with a 16 GB 3733 MHz LPDDR4X RAM on time scales ranging from a few seconds (e.g. evaluating SCE in Exp.2) up to approximately an hour (e.g. SCE-based learning in Exp.3). Our code is available at: `https://anonymous.4open.science/r/Structural-Causal-Explanations-D0E7/README.md`.

---

**Algorithm 1: Learning w/ SCE Regularizer**

---
**Input**: Data $\mathbf{D}$, SCM learner $\mathcal{M}_{\boldsymbol{\theta}}$, Optimizer $\mathcal{O}$
  **Output**: SCM $H$, Causal Graph $G$

**while** $i \leq |\mathbf{I}|$ **do**
  $H, l \leftarrow \mathcal{M}_{\boldsymbol{\theta}}(\mathbf{D})$  ▷ estimate SCM & loss
  $Q_X, \boldsymbol{I}^* \leftarrow \boldsymbol{I}_i$  ▷ sample Query-SCE pair
  $\boldsymbol{I} \leftarrow \mathbf{E}(Q_X, H, \mathbf{D})$  ▷ generate new SCE
  $l_{\boldsymbol{I}} \leftarrow ||\boldsymbol{I} - \boldsymbol{I}^*||_2^2$  ▷ compute penalty
  $\boldsymbol{\theta} \leftarrow \mathcal{O}(l + l_{\boldsymbol{I}}, \boldsymbol{\theta})$  ▷ parameter update
**end while**
$H \leftarrow \mathcal{M}_{\boldsymbol{\theta}}(\mathbf{D})$, and $G \leftarrow |\tanh(H)|$
**return** $H, G$

---

## A.3 THEORETICAL PROPERTIES OF *ER* AND SCE

The concepts of why-question, causal scenarios and *ERi* rulest hat we had to develop for the introduction of SCE algorithm, alongside SCE itself, come with several mathematical consequences which we now discuss. All of the subsequent results are simple and can be proven easily, still, their importance needs to be stressed since they make implications about the wide applicability of SCE.

**Proposition 2.** *For any causal scenario the rules ER*1 *and ER*2 *will be mutually exclusive.*

*Proof.* First, we code the binary ordering relations $<, >$ to represent 0 and 1 respectively. Second, we observe that ER$i \in \{<, >\}, i \in \{1, 2\}$ always involves the triplet $T = (R(\alpha, 0), R(v_j, \mu_j), R(v_i, \mu_i))$. Third, let $\mathbb{T} := \{0, 1\}^3$ be the set of all such triples as their code words, so $T \in \mathbb{T}$. Looking at the total number of possible scenarios $|\mathbb{T}| = 2^3 = 8$, we easily see that ER1 covers codewords $\{010, 011, 100, 101, 000, 111\}$ and ER2 covers the codewords $\{001, 110\}$, and together they cover all codewords ER1 $\cup$ ER2 $= \mathbb{T}$. Since any single scenario $C_{i,j}$ is uniquely mapped to a codeword, it will either trigger ER1 or ER2 but never both. □

**Proposition 3.** *The SCE recursion always terminates.*

*Proof.* The recursion's base case is reached when a root node is reached i.e., a node $i$ with $\mathrm{pa}_i = \emptyset$. An SCM implies a finite DAG, so root nodes are reached eventually. □

**Proposition 4.** *The output of any causal structure learning algorithm can be used to compute SCE.*

*Proof.* The proof for this proposition is surprisingly simple in that the SCM $\mathcal{M}$ used in the SCE recursion is only required to provide some kind of numerical value $\alpha$ for the relation of any variable pair $(V_i, V_j)$, that is, a matrix $\mathrm{A} \in \mathbb{R}^{|\mathbf{V}| \times |\mathbf{V}|}$ which represents a linear SCM or a SCM where each $\alpha$ represents a causal effect description. If the matrix $\mathrm{A}$ is an adjacency matrix living in $[0, 1]^{|\mathbf{V}| \times |\mathbf{V}|}$, then we simply have no information about *ER*3 since all causal effects are assumed to be the same. Since any causal structure learning algoirthm will produce a causal graph represented by a matrix, we have that we can compute SCE. □

The beauty of Prop.4 can be fully appreciated when being put into the context of practical AI/ML research and application. It tells us that *any* causal graph learner ever invented and that will ever be invented can provide causal explanations on any query of interest consistent with the learned model thus reflecting the learnt. In practice this means that all prominent graph learning algorithms like NT (Zheng et al., 2018), CGNN (Goudet et al., 2018), DAG-GNN (Yu et al., 2019) and NCM (Ke et al., 2019) are all explainable[1]. On a concluding note to this section, we have a remark on SCM

---

[1]The DAG learner in NT can be interpreted as a linear SCM but there is no guarantee.

that allow for hidden confounder. SCE as presented Def.7 do not cover hidden confounders and we leave this for future work. However, we can always modify the algorithm to talk about "unknown reasons" when giving knowledge on $\mathbf{U}$. An extended discussion on this and also other noteworthy aspects of SCE can be found in the Appendix.

## A.4  DETAILS FOR USING SCE AS REGULARIZER

We made use of NT as representative of graph learners for this experiment in which we investigate whether SCEs themselves can be used as a supervision signal to improve the quality of the learned graph. To circumvent the non-differentiable nature of our recursive formulation of SCE we train a neural network on a set of legal SCE to mimic the algorithm's output while being fully differentiable. Following Zheng et al. (2018), we generate 70 random linear SCMs following Erdos–Renyi structures. We use graph induction to infer 70 more graphs, making 140 in total. For each graph we generate 50 random why-questions to be answered, resulting in a data set of 7000 explanations. We extend the NT loss composition with this neural approximation using a SCE regularization penalty (to penalize SCE inconsistent graph estimates) and perform graph induction once with and once without the regularization (where between 1 and 50 explanations are being observed). The graph induction is being performed in a data-scarce setting with only 10 data samples per graph induction. Thus to infer the true causal structure the method ideally needs to perform sample-efficient. Main paper Fig.4 shows our empirical results on the error distributions for all the graphs while presenting the qualitative difference in the estimated graphs for the most significantly improved example. It can be observed that with the regularization the induction method can both identify more key structures while significantly reducing the number of false links, thereby appearing to be overall more sample-efficient. An explanation would be that, as conjectured, the explanations contain valuable information about the underlying SCM if the explanations themselves were generated by a similar SCM, thereby striking structures that would lead to contradicting explanations.

## A.5  DETAILS FOR FEEDING SCE WITH GRAPHS LEARNED FROM DATA

We select NT (Zheng et al., 2018) as a representative data-driven graph learner for the illustration in Fig.7 which considers the previously covered data sets and why-questions i.e., weather forecast (W, Mooij et al. (2016)), health (H), mileage (M), and recovery (R, Charig et al. (1986)). The SCE generated using the learned causal semantics are identical for the DW and M data sets, while differing only subtle for R and drastically for CH data sets. The former discrepancy occurs on the second-level of reasoning i.e., the right top-level explaining answer is given to the question (i.e., "Kurt did not recover because of the problematic pre-conditions") but was contrasted wrongly (i.e., the treatment countering the state of condition and not affecting the condition). The latter discrepancy revolves around a totally different structure e.g. the learned model expects a direct cause-effect relation between age and mobility while also wrongly assuming that food habits have a detrimental effect on health. An explanation in the case of NT is clearly the violation of the linearity assumption for the CH data set generating SCM.

While in Prop.4 we prove that graph learner are generally explainable in the sense of SCE, for empirical illustration we also provide more examples of such graph learner-based SCE, as we did with our lead examples for NT, in this case additionally for CGNN (Goudet et al., 2018) and DAG-GNN (Yu et al., 2019). Figure 7 and Table 2 show an application to NT with graph visualizations and of all methods to a superset of questions (that is, same and more) as the data used for NT. It is crucial to note that the presented results have *not* been hyperparameter-optimized (HO). Take for example CGNN, where candidate selection is exhaustive (brute force, and thus super-exponential in the number of nodes) and the model selection heavily relies on the neural approximation, thereby, HO is likely to be important. In a nutshell, the motivation behind Tab.2 is to present support for our theoretical proof on SCE-interpretability of graph learner i.e., we also give empirical proof for several methods in practice (opposed to pure theory). To assess the quality of the SCE, it is important to note the assumptions made by the original method. E.g., NT and DAG-GNN assume linear SCM. Thereby, we have *no guarantees* for running such a method in a non-linear data domain (which we do with the data sets DW and CHD). *Interestingly, these assumptions can in fact be exposed by SCE.* Consider the DW data set (Tab.2, first example), theory suggests that a linear model with Gaussian noise will exist in both directions $X \rightarrow Y$ and $Y \rightarrow X$, thus being non-identifiable (Peters et al., 2017). Methods like NT and DAG-GNN therefore pose the assumption that the given

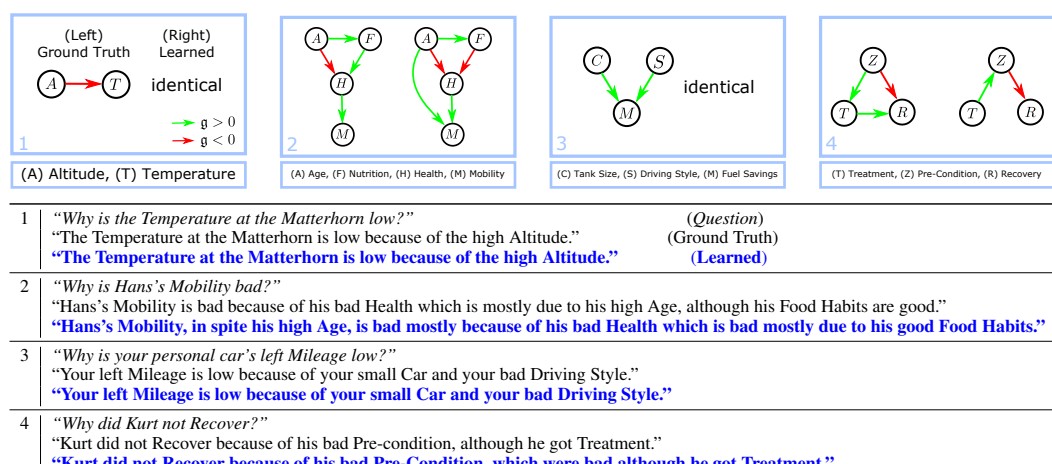

Figure 7: **Quality of Learned Interpretations.** We chose the simple, popular NT from Zheng et al. (2018) as our graph learner for generating the SCE. Subtle differences between explanations exist e.g., the explanation 4 is right on the top-level but for the wrong reasons, that is $T \to Z$ instead of $T \to R$. Variable letters are capitalized. (Best viewed in color.)

data comes, in this case, from a linear model with Gaussian noise i.e., the identifiability problem is being circumvented altogether. This is also the reason why different random seeds can lead to both modellings ($A \to T$ and $T \to A$) for the DW data set (see in Tab.2 how the SCE flips for $\mathcal{M}_3$=DAG-GNN for the two opposing DW queries). Another important note is that the uninformed SCEs "No causal explanation ..." occur when the method's SCM estimate does not contain a causal path to the variable that is being queried by the why-question i.e., the SCM will actually contain a non-trivial estimate of the underlying causal structure, even though the SCE returns a trivial/empty explanation since the variable of interest can not be reached within the estimate's structure with a directed path (i.e., the base case in Def.7 is trivially triggered). In fact, these negative "no answer"-type of cases are important since the model need also be able to know when there is nothing to be known. For this case, we also pose why-questions to which the ground truth is already a "no answer" explanation since there is no causal connection to the variable being queried by the why-question. The empirics in Tab.2 suggest, as theoretically proven (Prop.4), that the graph learner are explainable and also that all 3 rules (excitation, inhibition and preference) are being used for the graph learner-based SCE. As a positive example, consider example #3 for the CH data set where $\mathcal{M}_1$ captures the complex explanation correctly up to preference and falsely assuming that food habits ($F$) have a negative causal effect on health ($H$). A more interesting example (#8 for the R data set) shows that the main reason being bad pre-conditions ($Z$) is being captured but the model falsely assumes that those are because of the received treatment ($T$). To consider a negative example have a look at example #4 again for CH where the actual answer is a "no causal explanation" since age ($A$) is a root node. However, $\mathcal{M}_3$ claims that the age is high because of the food habits and mobility ($M$), then again because of health. While the statement is wrong and also feels exaggerated, inspecting closely one can detect the correct existence of the causal edge between mobility and health ($H \to M$). I.e., the model interprets wrongly, but its causal model is still partially valid.

## A.6 DETAILS FOR USER STUDY

We instructed $N = 22$ participants to answer our questionnaire (see appendix Fig.9). The questionnaire asked the following questions: *"Given a pair of variables, does a causal relationship exist (existence)? If yes, then which is the cause and which is the effect (direction)? If there are multiple causes for a single variable, then how impactful is each of the causes (preference)?"* All of these questions, alongside their responses, are of *qualitative and subjective* nature. It is important to note that the participants *do not* perform the actual induction from specific, provided data like the algorithms do i.e., the human subjects are not given the variable names nor concrete data points that would allow them to find the rules for the specific data sets. Instead, they were only given the variable names/depictions, thereby having to induct from personal experience/understanding essentially.

This approach to human induction is related to the experimental setups in (Griffiths & Tenenbaum, 2006; Hattori, 2016).

The motivating lead research questions we intended to answer, and in fact do answer successfully with this experiment, are: What are SCM that (some) human could model? How does overlap for human-based SCM occur? How do subsequent SCE (Def.7) between humans and algorithms differ? In a nutshell, we wanted to investigate the similarity of SCMs between subjects in addition to the similarity between subjects- and algorithm-based SCEs.

A caveat regarding the analysis and explanation of human judgements is that sample bias may distort conclusions. Sample bias has long been identified within the behavioral and social sciences as limiting the generalization of results obtained in a specific sample to the population. A common methodological fix to counteract such biases is to increase the sample size, see (Daniel, 2017) for a recent application and discussion. Certainly, the observed sample will affect the way the difference (to e.g. algorithm-based SCE) turns out to be, but then again our research question is *not* concerned with all possible human explanations, but any. Furthermore, we chose data sets that model very general examples and thus offer accessibility to the general population since no single person might be an expert. Ultimately, this way of designing our experiment, while not removing sample bias of course, renders the bias's qualitative effect onto our subsequent investigation negligible.

In the following we provide a discussion of several interesting and important insights discovered through the human user study. Nonetheless, it is important to note that our results like most modern day interpretations of human behavior are of conjectural nature – sensible, educated guesses essentially. During this discussion, we will point to specific aspects of the descriptive statistics displayed in appendix Fig.10. The actual human data is also being appended for the sake of completion (click on the following link to access the anonymized human data: `https://anonymous.4open.science/r/Structural-Causal-Explanations-D0E7/Survey-Human-Data-Anonymized.pdf`). The questionnaire contains four examples with two, three, three, and four variables (or concepts) respectively that are being visually depicted in addition to a concise textual description. We randomized the textual description of up to three variables across all examples for any randomly selected participant. Doing so, we allow for the randomized concept to reverse causal influence directions, thus, diminishing the bias of chance-selecting said causal direction – in a nutshell, this randomization scheme helps us in controlling for explanation variance (or leeway) of the subjects. Nonetheless, we still observed that for any variable pair $(X, Y)$ the meanings of $X$ and $Y$ themselves could be interpreted differently, which ultimately resulted in False Negatives regarding agreement i.e., people will disagree technically although they actually agree. To give a concrete example, consider the following: pre-condition in Example 2 can be interpreted as "the length of the medical history of a patient" (negative; increasing implies lower chance of recovery) opposed to "the state of well-being of a patient" (positive; increasing implies higher chance of recovery), thereby some subjects might choose $Z_1 \rightarrow R$ while others will choose $Z_2 \leftarrow R$ where $Z_i$ are the different explanations of the "pre-condition" concept (and $R$ denotes recovery), yet all subjects agree on an existing relation between the two variables: $Z_i \leftrightarrow R$. Also, some variables/concepts were more stable in their explanation variance. To give yet another specific example, altitude and temperature in Example 1 (appendix Fig.9) are stable concepts while the aforementioned pre-condition in Example 2 is unstable (due to its explanation variance/leeway). More importantly these different explanations due to the ambiguity inherent in language become visible within the statistics. To stay inline with the previous example, consider the medical example within appendix Fig.10 (second row, middle) and specifically consider the edges $T \rightarrow R$ and $Z \rightarrow R$. For the former relation the agreement between subjects is evident i.e., the majority of human subjects will select this edge. For the latter relation, we clearly see the two previously discussed explanations that subjects employ during edge decision. I.e., for some subjects the edge between $Z$ and $R$ is positive and for some others it is negative, while naturally all agree upon there being a relation between the variable pair ($Z \leftrightarrow R$) opposed to there being no relation ($Z \nleftrightarrow R$).

We observe a systematic approach and thereby non-random approach to edge-/structure-selection by the human operators, see any of the subplots within appendix Fig.10. Furthermore, there are only a few clusters even with increasing hypothesis space. Both the systematic manner and the tendency to common ground are evidence in support of the MMC hypothesis (MM $\equiv$ SCM, Hyp.1) and its implied argument on "true" SCM information reachable from the overlapping MM-based SCMs or SCMs.

Although we randomize the order of variables in addition to consistently presenting them in a simple line with the intention of not inducing any specific sorting/structure to avoid bias, we still observed apparent, unintended subject behavior. For instance, subject number 5 only considered pairs presented next to each other as being questioned although the other combinations are meant to be queried as well. While additional research needs to corroborate these observations, our data suggests that attention might have decreased over the course of the experiment for a subset of subjects as suggested by e.g. subject number 7 where overall agreement with the subject majority is to be found but eventually at the very last example "mistakes" occur (specifically, the subject highlighted that "increasing age increases mobility", in stark disagreement with the majority of participants). We also observe that the increase in hypothesis/search space (i.e., more variables) comes with an increase in variance. This variance increase can be argued to be due to the progressive difficulty of inference problems as well as decreased levels of attention and potential fatigue across the duration of the experiment (e.g. consider the duplicate plots, third column, in appendix Fig.10 where the number of unique structures that are being identified increases significantly). Yet another interesting observation concerns the aspect of time, consider subject number 17 where there is a cycle between treatment and recovery where the subject likely thought in terms of "increasing treatment increases speed of recovery *which subsequently* feeds back into a decrease of treatment (since the individual is better off than before)" which seems like a valid inference but clearly considers the arrow of time. Yet another observation, some subjects faced questions of variable scope e.g. if there is a causal connection between food habits and mobility, then some subjects considered energy as the mediator and since energy is not part of the variable scope, confusion might arise whether to place an edge between food habits and mobility or not. In fact, for such a scenario the correct answer is to place an edge, since there exists a causal path from food habits to mobility, via energy, even if energy is not displayed. I.e., in causality, an edge can/will talk implicitly about all the more fine-grained variables that are part of the causal edge/path.

The second data set is an instance of the famous Kidney Stone example (Peters et al., 2017), where $Z$ is a confounder that indicates the pre-conditions in terms of e.g. the size of the kidney stone, and it also illustrates the famous Simpson paradox (Simpson, 1951; Pearl, 2009; Peters et al., 2017) where the recovery will favor one treatment in the overall statistics while being better for all of the non-consolidated views for the other treatment. We observe that not a single subject places the edge pre-condition to treatment ($Z \rightarrow T$) which is arguably at the core of Simpson's paradox. This observation gives an additional cue on why the phenomenon is called paradox because no human subject expects the existence of this connection and even actively neglect the existence.

We observe that the human-based SCE match the Ground Truth SCE *perfectly* up to the R data set SCE, which is also the "Result" in Fig.4 i.e., the "Mode" approach returns the correct SCE while the "Greedy" approach chooses the wrong edge type for $Z$ and $R$. After further investigation, we believe to have found several explanations for this "human" mistake that we discuss extensively in the appendix. On another note, we observe that the overall flawless performance of human-based SCE speaks for superiority over algorithmic graph learner-based SCE. To conclude this paragraph, let us appreciate one such drastic difference in explanations, which in fact occurred on our lead example "Causal Hans":

> **Humans**: *"Hans's Mobility is bad because of his bad Health which is mostly due to his high Age, although his Food Habits are good."*
> **Machines**: *"Hans's Mobility, in spite his high Age, is bad mostly because of his bad Health which is bad mostly due to his good Food Habits."*

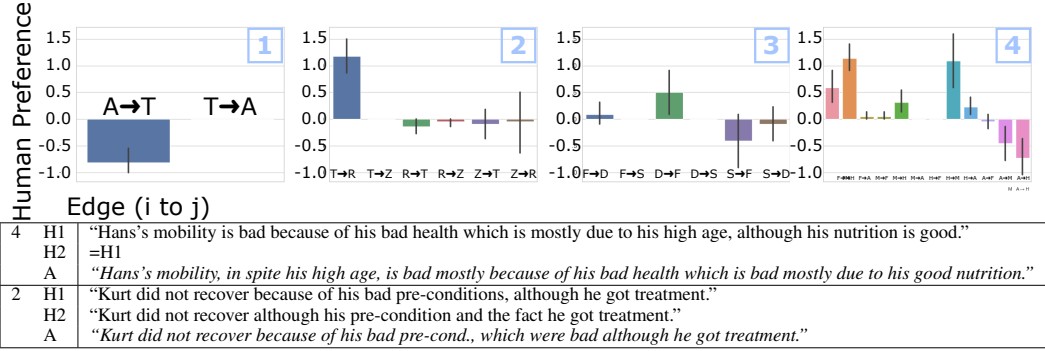

| 4 | H1 | "Hans's mobility is bad because of his bad health which is mostly due to his high age, although his nutrition is good." |
| | H2 | =H1 |
| | A | *"Hans's mobility, in spite his high age, is bad mostly because of his bad health which is bad mostly due to his good nutrition."* |
| 2 | H1 | "Kurt did not recover because of his bad pre-conditions, although he got treatment." |
| | H2 | "Kurt did not recover although his pre-condition and the fact he got treatment." |
| | A | *"Kurt did not recover because of his bad pre-cond., which were bad although he got treatment."* |

Figure 8: **"Humans vs Machines".** Top: Edge plots per example where the bars denote the average value of given relation and the errors confidence intervals. Bottom: The SCE generated for the two human variants against a graph learner. Human explanations are (near-)identical to the ground truth from Tab.7. (Best viewed in color.)

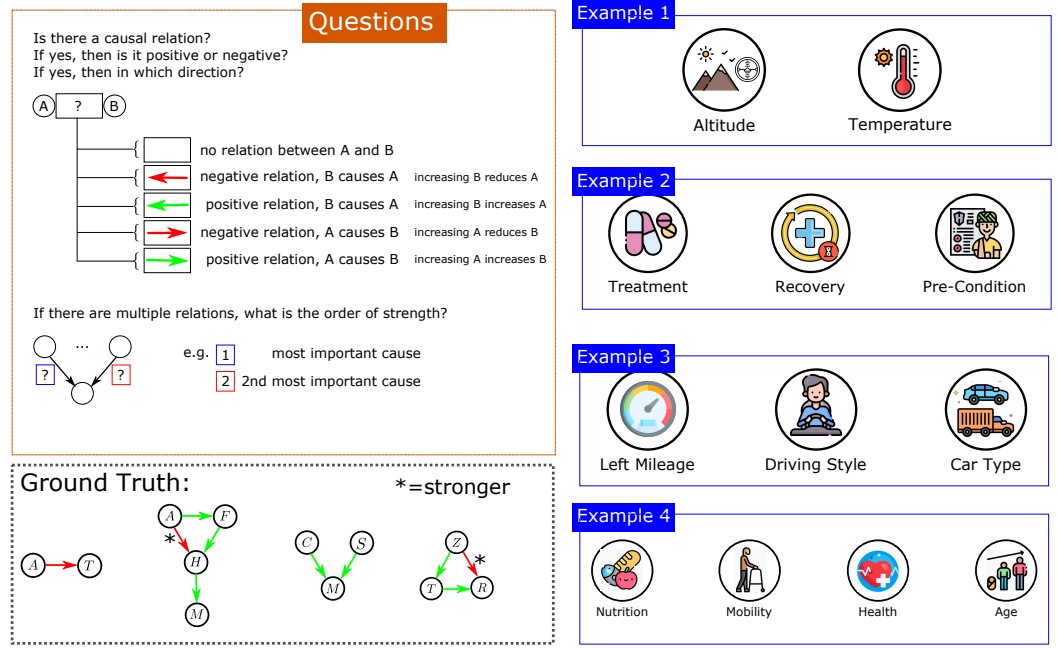

Figure 9: **Experiment Setup for the Human Case Study.** The participants are being asked two questions: whether there is a directed relation between some variable pair $A$ and $B$, and when there are multiple causes how they behave relatively i.e., the order of strength in relations. We avoid bias in drawing relations by randomizing the order and presenting the variables in a sequence. Induction is being performed from personal "data"/experience, rather than by looking at a matrix of data points. To avoid bias in drawing relations, we don't provide any hints on a graph structure and we randomize the sorting of the variables. To provide more clarity we depict the names of the concepts with additional illustrations. The participants are asked to perform induction based on personal data/experience i.e., they only see the orange and blue boxes. (Best viewed in color.)

| | Structural Causal Explanation (Prop.4) |
|---|---|
| #1 | Dataset: DW, Query: "Why is the temperature at the Matterhorn low?" |
| GT | "The temperature at the Matterhorn is low because of the high altitude." |
| $\mathcal{M}_1$ | "The temperature at the Matterhorn is low because of the high altitude." |
| $\mathcal{M}_2$ | "The temperature at the Matterhorn is low because of the high altitude." |
| $\mathcal{M}_3$ | "No causal explanation for Matterhorn's temperature." |
| #2 | Dataset: DW, Query: "Why is the Matterhorn so high?" |
| GT | "No causal explanation for Matterhorn's altitude." |
| $\mathcal{M}_1$ | "No causal explanation for Matterhorn's altitude." |
| $\mathcal{M}_2$ | "No causal explanation for Matterhorn's altitude." |
| $\mathcal{M}_3$ | "The altitude of the Matterhorn is high because of the low temperature." |
| #3 | Dataset: CH, Query: "Why is Hans's mobility bad?" |
| GT | "Hans's mobility is bad because of his bad health which is mostly due to his high age, although his nutrition is good." |
| $\mathcal{M}_1$ | "Hans's mobility is bad because of his bad health which is bad because of high age and mostly due to his good nutrition." |
| $\mathcal{M}_2$ | "Hans's mobility is bad because of his good nutrition." |
| $\mathcal{M}_3$ | "No causal explanation for Hans's bad mobility." |
| #4 | Dataset: CH, Query: "Why is Hans old?" |
| GT | "No causal explanation for Hans being old." |
| $\mathcal{M}_1$ | "No causal explanation for Hans being old." |
| $\mathcal{M}_2$ | "No causal explanation for Hans being old." |
| $\mathcal{M}_3$ | "Hans is old because of his good nutrition and bad mobility, which is because of his bad health." |
| #5 | Dataset: CH, Query: "Why is Hans's nutrition good?" |
| GT | "Hans's nutrition is good because of being older." |
| $\mathcal{M}_1$ | "Hans's nutrition is good because of being older." |
| $\mathcal{M}_2$ | "No causal explanation for Hans's nutrition." |
| $\mathcal{M}_3$ | "Hans's nutrition is good because of his bad health and mobility." |
| #6 | Dataset: M, Query: "Why is your personal car's left mileage low?" |
| GT | "Your left mileage is low because of your small car and your bad driving style." |
| $\mathcal{M}_1$ | "Your left mileage is low mostly because of your small car and because of your bad driving style." |
| $\mathcal{M}_2$ | "No causal explanation for the left mileage." |
| $\mathcal{M}_3$ | "Your left mileage is low because of your small car and your bad driving style." |
| #7 | Dataset: M, Query: "Why is your personal car small?" |
| GT | "No causal explanation for the car size." |
| $\mathcal{M}_1$ | "No causal explanation for the car size." |
| $\mathcal{M}_2$ | "Your personal car's size is small because of your good driving style and fuel savings." |
| $\mathcal{M}_3$ | "No causal explanation for the car size." |
| #8 | Dataset: R, Query: "Why did Kurt not recover?" |
| GT | "Kurt did mostly not recover because of his bad pre-conditions, although he got treatment." |
| $\mathcal{M}_1$ | "Kurt did not recover because of his bad pre-conditions which is because of the treatment he got." |
| $\mathcal{M}_2$ | "No causal explanation for Kurt's recovery." |
| $\mathcal{M}_3$ | "No causal explanation for Kurt's recovery." |
| #9 | Dataset: R, Query: "Why did Kurt get treatment?" |
| GT | "Kurt got treatment because of his bad pre-conditions." |
| $\mathcal{M}_1$ | "No causal explanation for Kurt's received treatment." |
| $\mathcal{M}_2$ | "Kurt got treatment because of his bad pre-conditions." |
| $\mathcal{M}_3$ | "No causal explanation for Kurt's received treatment." |

Table 2: **SCE Evaluation on Graphs Learned from Data.** We prove Prop.4 for general graph learners while pointing to some example methods from the existing literature on graph learners. Here we show the results of running the methods $\mathcal{M}_i$, 1:NT (Zheng et al., 2018), 2:CGNN (Goudet et al., 2018), 3:DAG-GNN (Yu et al., 2019) on the four data sets weather forecast (W, Mooij et al. (2016)), health (H), mileage (M), and recovery (R, Charig et al. (1986)) for the respective why-questions. As suggested, the methods are explainable and reveal insights onto the learned causal semantics, while varying significantly in quality in terms of accuracy relative to the ground truth (GT). Independent of accuracy, "No causal explanation ..." occur when the SCM estimate of $\mathcal{M}_i$ contains no causal path to the queried variable $X$ i.e., $\text{pa}_X = \emptyset$ (supported through GT sparsity). We also show GT explanations that require a negative "no answer" response by $\mathcal{M}_i$.

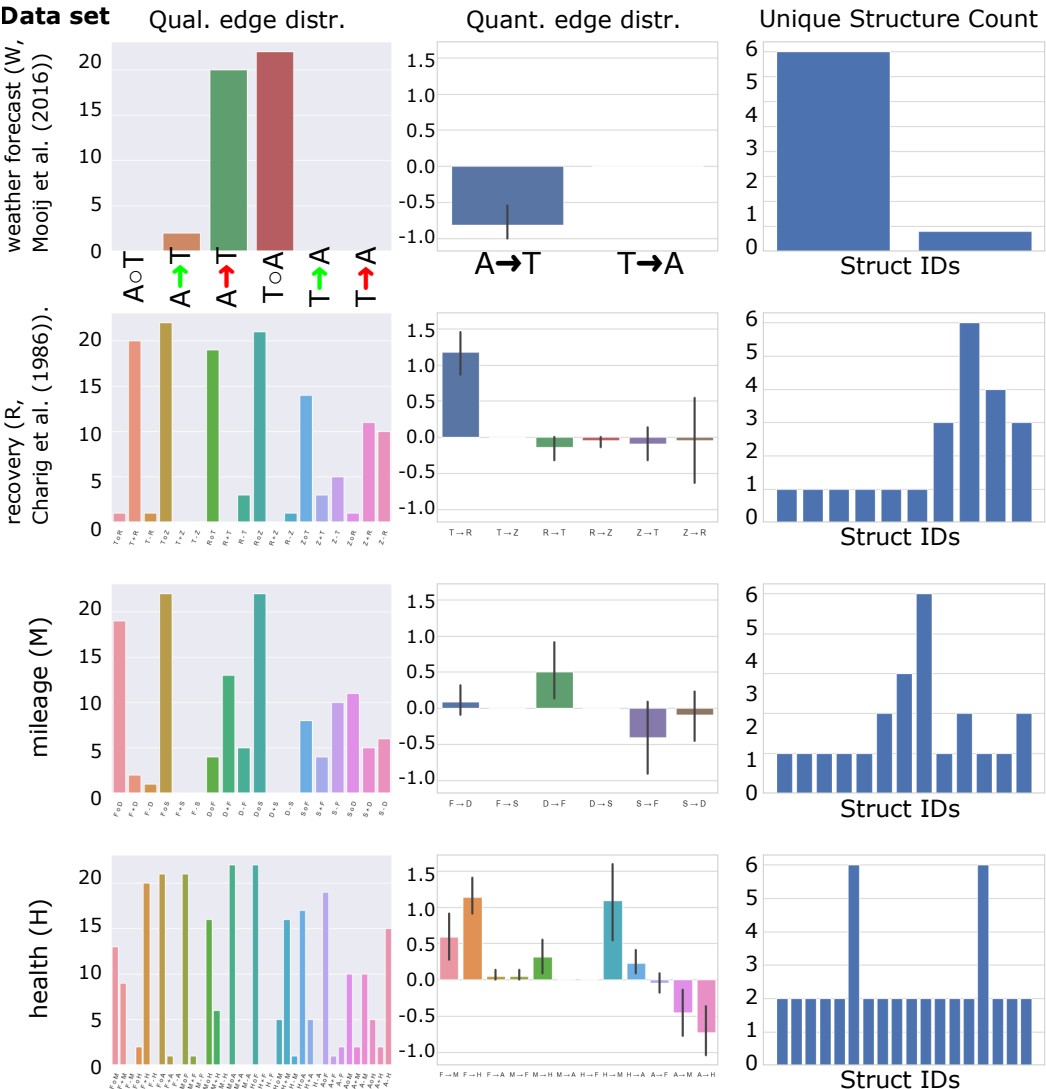

Figure 10: **Human Data Analysis: Qualitative, Quantitative, and Uniqueness.** Statistics collected from the human data ($N = 22$). The rows denote the four data sets: weather forecast (W, Mooij et al. (2016)), recovery (R, Charig et al. (1986)), mileage (M), and our synthetic health data set (H). The columns: qualitative edge distributions that show for each of the different edge type how often it was chosen respectively (left). Qualitative because it incorporates ACEs. A green edge denotes positive ACE, whereas red means negative ACE (i.e., increasing $X$ would decrease $Y$). Quantitative edge distribution for each edge where the error bars denote confidence intervals (middle), and the unique structure counts where each bar depicts the frequency of a qualitative structure discovered by the human subjects (right). Extensive elaboration on the setup, execution and results of this human study are to be found in the corresponding appendix section. (Best viewed in color. Since the plots get dense with increasing combinations of variable pairs, please consider zooming in to read the labels for any detailed view of the results.)

