# OpenReview forum: "Generating Explanations From Linear Structural Causal Models"
_ICLR.cc/2024/Conference — ICLR 2024 Conference Withdrawn Submission_

### Official Review · Reviewer_bZoD · 2023-10-31

**Soundness:** 3 good
**Presentation:** 3 good
**Contribution:** 1 poor
**Rating:** 3
**Confidence:** 4

**Summary:**

Given a linear structural causal model (SCM), this paper presents a technique for deriving a 'structural causal explanation' (SCE) like: "Hans' Mobility (M) is bad because of his bad Health (H), which is mostly due to his high Age (A), although his Nutrition (N) is good".

The technique is empirically tested by asking $N = 22$ human subjects whether pairs of variables are causally related, and to rank the related pairs by the strength of the relationship.  The experiment shows the SCEs generally lie close to human answers.

**Strengths:**

**originality**

I have not seen this analysis before.

**quality**

The paper's steps seem properly conducted.

**clarity**

The paper is generally well explained.

**significance**

**Weaknesses:**

1. the abstract claims that generating explanations from causal representations "has not been studied before''.  (This is always a risky claim to make!)  Frye et al. (NeurIPS 2020) comes to mind as a possible counter-example: it showed how to use causal knowledge derived from e.g. a SCM, to augment Shapley value explanations.  If the authors object that they want 'textual' explanations, then it would be trivial to extend Frye et al. to produce statements like "Hans' Mobility (M) is bad because of his bad Health (H), which is mostly due to his {whatever feature has largest causal Shapley value}".  (I have read Heskes et al. before, which this paper cites, but do not recall whether its approach is closer to the current approach.)

1. the example against which the current work is motivated uses a technique called CXPlain, which is both not explained, and which I had not heard of.  Thus, this struck me as a weak motivation.  By contrast, Rudin's work on simple textual explanations comes to mind as an example of good motivations.

1. in linear models, the problem of explainability is less than in more general models, to the point where it is not clear to me that there is a problem: the magnitude of the coefficients on a linear regression in which features are deviated from their means should allow production of statements like the "Hans' Mobility (M) is bad because..." statement.  Thus, I would want to see the value-added of the approach presented here against a simple linear regression.  (Yes, the linear regression cannot learn an SCM, but the paper assumes that "we can realistically expect to have access to an estimate of" the SCMs.)

1. ultimately, it seems to me that the paper is built around a fairly simple technique, but is then complicated by "inspiration from neuroscience", recursion, FOL, appendix material on mental models, etc.  I would recommend presenting the simple idea cleanly, and then properly substantiating it (by comparison to alternative approaches), rather than complicating it.  The result, here, compared to something like Frye et al.'s causal Shapley value, feels much less transparent.

1. the empirical study is unconvincing: a small number of people seem to have been asked some fairly easy questions; the SCE captures those responses.  A central element of Imbens' critique of Pearl ("Potential Outcome and Directed Acyclic Graph Approaches to Causality") is that it does not scale: the 'toy' nature of the examples presented here do not overcome that concern.  As I suspect that nothing computational prevents scaling, it would be nice to see a proper example.

1. p.9 "Shapley" is misspelled as "Shapely" in Heskes et al.

**Questions:**

1. given that Frye et al.'s causal Shapley values can handle non-linear relationships, what is the relationship between it and the current SCE in linear environments?

1. the SCEs presented seem scale dependent: q.v. p. 5: "$\beta > \gamma$ means that the causal effect of aging on health is greater in absolute terms than the causal effect of nutrition on health".  Will changing the units in which the two causal variables are labelled also change the explanation?

1. in Figure 2, should 'N' or 'P' be used for 'Nutrition'?  'N' has been used in the preceding.

---

### Official Review · Reviewer_iBei · 2023-10-31

**Soundness:** 2 fair
**Presentation:** 3 good
**Contribution:** 3 good
**Rating:** 3
**Confidence:** 2

**Summary:**

The paper offers an automated way to generate causal explanations (formalized as SCEs or structural causal explanations). These are qualitative descriptions of how various factors affect an observed result, answering the question "why?" It's shown empirically that regularizing with SCEs can potentially improve graph learning and that they accord with human intuitions.

**Strengths:**

The problem is interesting and the approach that they suggest is interesting and novel to my knowledge.
The writing is clear overall and the concept is analyzed from various angles including empirical.

**Weaknesses:**

There seems to be a fundamental issue with the approach here. Namely, while (as the authors note) explanations belong to the counterfactual level, everything that is actually done with SCEs here seems to be in terms of causal effects. This problem cannot be hand-waved away since as Barenboim, Correa, Ibeling, & Icard 2020 ("On Pearl's Hierarchy...") and Ibeling & Icard 2021 ("Topological Perspective on Causal Inference") have shown, counterfactuals can almost-never be determined from causal effects. It is likely that they are (subjectively) "substantially" underdetermined in a subspace of large measure.
In Example 14 in the Technical Report R-60 version of the former paper, there is already an example showing how considering counterfactuals can flip qualitative comparisons between different quantities (<, >). This seems particularly pertinent to SCEs and this example can probably be massaged into one in which the SCEs generated are actually unsound at the counterfactual level.
It is possible that I misunderstood—if so, I invite the authors to show how their approach actually operates with counterfactuals rather than merely with causal effects. If not then I think this may unfortunately mean that what their approach generates are not explanations in the true, counterfactual sense, substantially weakening the contribution.

**Questions:**

See weakness above.
Please give the runtime of the SCE-generating algorithm.
Minor comments:
- please expand acronyms and abbreviations ("sth" or "w.r.t." and so on) except for standard ones
- it is clearer to mark the algorithm as a separate environment box, e.g. "Algorithm 1"

---

### Official Review · Reviewer_7kYr · 2023-11-05

**Soundness:** 3 good
**Presentation:** 2 fair
**Contribution:** 2 fair
**Rating:** 3
**Confidence:** 3

**Summary:**

The paper presents a formalisation of explanations based on linear structural causal mechanisms, and deploy a simple recursive algorithm to generate such explanations (i.e., SCEs). Moreover, the paper carries out empirical studies with human participants, using a common causal discovery algorithms, confirming that when incorporated via naive regulariser (that they propose) their SCEs are indeed helpful.

**Strengths:**

-There is originality in the paper in defining the explanations formally. Interesting work.

- Formal maturity and quality is above average (although not clear all the time, see weaknesses).  The paper takes its time in explaining the introduced notions and examples,  and making discussions.

- Carrying out empirical studies through humans when it comes to explanations is a relevant effort.

**Weaknesses:**

-The paper depends too much on appendix in many details. This includes all the theoretical results and the experimental  results.  It's hard to assess the soundness of it without going through details in appendix  (hence my confidence score) which is longer than the actual paper. And In its current form,  I can't see how it is helpful as a publication in conference proceedings, provided that it is accepted. I would encourage authors to rather submit to a journal.

- Related to above the   tries do so much at once and spreads thin in the main text and spreads thin. I would recommend authors to decompose paper  to either theory or experimental section, or the incorporating the SCEs to learning algorithm and present the results deeply in the main text.  Move some of these contributions to main text and in places.

-Although having good mathematical maturity (yet not clear all the time, see below), paper presentation at times are too wordy. In particular,  the paper presents some technical observations and totally skips the details, while it is too wordy when it comes to conceptual explanations and discussions, while being dry.  I would suggest authors to make bullet lists for many inline texts.


- The claim in abstract  " Surprisingly, this conceptual idea of generating explanations mainly from a suitable causal representation, like Pearl’s Structural Causal Model (SCM), has not been studied before." is a big one, and can be misinterpreted and easy to disagree. I recommend them to flourish it and be more specific.  I would recommend to cite  "Causal Explanations and XAI 2022" by Sander Beckers in introduction or related work.

- Definition 6, First Order Logical formalisation is not clean, and obscure.  If you are proposing and pronouncing it, then it needs to be full-fledged clear e.g., what are your predicates, what are your models,  what can we derive from them or prove are not clear.

- I appreciate the effort of pronouncing scheme the causal explanations from the formula with these sentences  but overall not fully convinced with the way it is presented to subjects, it needs further explanation/justification.

 In its current form, it's hard to pinpoint the depth and amount of contribution. Hence my score.



More minor issues:

- In the introduction,  " Suggesting that the structural causal model (SCM) representation.. " is not a sentence, because it is dependent on. the previous sentence.

-typo in abstract: defficiencies

-Desideratum 1 vs D2 , D3...

- in Preliminaries smaller v is not defined.

- Concerning,  Assumption 2.  you don't say anything about the independence of the exogenous variables. Why you  prefer to use  strict subset is not clear to me.

- Please spell out Causal Effect (CE), Average Causal Effect (ACE), Random variable (RV) etc.

- Shorthand notation  (Vi, V′) ⊂ V  is confusing, as the first reads as a pair.

- Definition 3, D comes first C second.

-Typo in conditional expectation for the continuous variable.

-Why does observation 1 holds needs a bit elaboration.

- why on " ..will simplify computation immensely..." is not clear, please explain.

- "with γ > 0 we expect the below average mobility" , gamma --> delta

-  "β > γ means that the causal effect of aging on health is greater in absolute terms than the causal effect of nutrition onto health." I assume  such effects are normalised right? otherwise it is misleading to say that.

- A Diagrammatic example  would serve better when it comes to A H N M with alpha gamma beta delta.

- Reconstructing the Causal Hans Example using Def.7: Better use a tree explaining the recursion.

- Too much technical detail on No Tears, looks quite random, and distracts the flow. Not clear why do you need this.

- typo:  "make make"

-typo in conclusion difficult[y]

**Questions:**

I don't have questions.

---

### Official Review · Reviewer_85pJ · 2023-11-06

**Soundness:** 2 fair
**Presentation:** 3 good
**Contribution:** 2 fair
**Rating:** 5
**Confidence:** 2

**Summary:**

This study is a novel approach that leverages SCM to generate explanations. While this concept of deriving explanations from causal representations is unexplored, the authors developed the first algorithm, primarily focusing on linear SCMs. They discuss the desiderata for this approach and address limitations in previous causal explanation methods. In an empirical study, the results present that SCE is a promising explanation scheme. They also propose a straightforward regularizer that incorporates SCE into the learning process. This study underscores the potential of integrating causal information into explanations and highlights limitations in existing graph learning algorithms.

**Strengths:**

- The study introduces a conceptually innovative approach to causal explanations, grounded in SCMs. This novel perspective offers a fresh angle on how explanations can be derived from causal structures, setting it apart from previous methods.
- The authors conducted an empirical investigation to assess the quality of SCEs and their integration with learning. The positive outcomes of this empirical section demonstrate the practical applicability and effectiveness of SCEs.
- This paper is well-organized and well-expressed.

**Weaknesses:**

- I am still confused by the results shown in this paper. According to my understanding, for linear SEMs, lots of causal discovery methods such as NOTEARS, GOLEM, and LiNGAM can identify the ground-truth DAG under some mild constraints. Why is it necessary to use this method to help give a further explanation?
- There is no theoretical guarantee that the causal explanation reveals the true causal relationships.

**Questions:**

- In definition 1, the definition of the parental set of $V_i$ seems incorrect in the third item, where $X_j <\mathbf{X} \ V_i$ means the ancestral set of $V_i$ instead of the parental set.
- There is no "independent exogenous variables" assumption in this paper, which is quite unusual. Where does the dependence come from?
- In the definition of ACE, The ACE set includes the causal effect of exogenous variables, which is not ready to be counted since we cannot adjust its value. It is better to replace $X$ and $Y$ with $V_i$ and $V_j$ in  Section 2.
- What does FOL mean?

---

### Official Review · Reviewer_hAzn · 2023-11-10

**Soundness:** 2 fair
**Presentation:** 2 fair
**Contribution:** 2 fair
**Rating:** 3
**Confidence:** 4

**Summary:**

The paper introduce a method for answering "Why?" questions in linear, Gaussian-noise, Markovian structural causal models called Structural Causal Explanations (SCE) and claims three contributions beyond this method:
1. "we conduct a human study to assess that SCE is sensible across various examples"
2. "we feed SCE with causal representations learned from data to assess what the explanations reveal about the underlying graph learning methods"
3. "we present a naive regularization penalty to reduce the number of false links in learned graphs"

The authors compare their method against CXPlain (Schwab & Karlen (2019)) as a benchmark, arguing their method satisfies three desiderata, which are deficiencies in CXPlain. Namely, they argue SCE can:
1. differentiate direct from indirect causes
2. capture qualtitative information on causal effects
3. cope with stochasticity

**Strengths:**

- Originality: this is the first natural-language-based explanation generated by a data structure derived from a structural causal model, to my knowledge.
- Significance: explanations are a highly relevant topic to the ICLR community.

**Weaknesses:**

Examining the construction of SCE in Definition 6 and Table 1, it seems that SCE explanations can indeed be expressed using natural language, but these expressions may be misleading and may not align with what we intuitively consider an explanation. Further refinement of the method to capture the cases described below where variables have more than one parent would be desirable. If these kinds of cases are not captured, SCE can only work in general in chain graphs of the form $X_1 \to X_2 \to X_3 \to \dots \to X_n$.

- ER3 in Table 1 doesn't seem to correspond to the phrase "Y is mostly because of [...]"; it seems that it would be more accurate to say "[...] contributes the most to Y", and this may not constitute an explanation. For instance, it seems wrong to say "The amount of money a charity received was high mostly because of Alfred, who donated 2 dollars" while 500 other people each donated 1 dollar, representing the amount of money donated as a variable $Y$, and the amounts of money donated by Alfred and each of the other 500 people as variables $X_1 \dots X_{501}$.
- ER3 doesn't seem to interact well with ER2. "Y is mostly although [X] is [low/high]" doesn't seem to have a meaning. For example, if a business has two sources of expense $X_1, X_2$ with weights $-1$ each and one source of income $X_3$ with weight $1.2$, the question "Why is revenue ($Y = - X_1 - X_2 + 1.2 X_3 = -1 -1 + 1.2 = -.8$) low?" would be answered with "Revenue is low is mostly although the income source is high." The grammar in this explanation isn't quite right, and while the proposed explanation would be true if "mostly" were removed, it does not tell me why the revenue is low. A better explanation would be "Revenue is low, because expense 1 and expense 2 are high."
- ER3 takes an argmax, preferring small positive weights over large negative weights. For example, let business expenses $E$ are very low and business income $I$ is moderately high, but revenue $R$ is low, and assume expenses have a negative effect on revenue while income has a positive effect. Then, the generated explanation would always be "Revenue is low mostly although income is high." It seems it would be preferable to explain the low revenue with the high expenses in this scenario.
- As an related note, it seems that explanations can become prohibitively long. For example, consider $X_1 \to \dots \to X_{500}$. The explanation for $X_{500}$ would necessarily contain 500 clauses, which a human would likely struggle to parse. Thus, even in the chain graph, a human might struggle.

Next, I examine the other individual contributions listed in the paper.

# Contribution 1: "we conduct a human study to assess that SCE is sensible across various examples"

Key details of the set-up of the study are left out, such as how coefficients or "high"/"low" status are inferred to generate SCEs. It is unclear which tables, figures, or data collected in the Appendix correspond to Observations (i-iv); it is unclear how a reader is supposed to verify these "Observations", which appear to in fact be claims or arguments. Therefore, I recommend that any key pieces of evidence in the Appendix be used to support the description of the user study in the main body of the paper, to improve clarity. From what evidence in the Appendix can I infer that SCE is 'sensible' across various examples, as stated in Observation (i)? How were the apriori expectations for explanations generated? Is the claim about being close to expectations statistically significant? How is closeness measured? Is there a baseline that SCE is compared to? All of these questions are important questions that I would recommend be answered in the main text of the paper.

# Contribution 2: "we feed SCE with causal representations learned from data to assess what the explanations reveal about the underlying graph learning methods"

What do the explanations reveal about underlying graph learning methods? What is the conclusion of your assessment? I didn't quite understand this from reading the section in pages 7-8. I would argue the conclusion of the assessment has the potential to be a contribution, rather than the assessment itself, and I would recommend explicitly stating any novel/significant conclusions in the contribution statement itself.

# Contribution 3: "we present a naive regularization penalty to reduce the number of false links in learned graphs"

The naive regularization penalty depends on having SCEs, which in turn depends on having the full graph. What is the significance or benefit of introducing such a penalty, given that it seems to require the graph as input, when the goal is to learn the graph? It seems that the empirical result is not significant, and I would recommend running more simulated trials so that the result does become significant. If it does not become significant with more trials, I would recommend not including it in the work, as it is unclear what is being shown if the claim in Fig 4 "Graph Learning Improves with Explanations" cannot be made with statistical significance.

# Note on related work

A comparison is made to CXPlain in the beginning of the work, with the primary criticisms being the desiderata mentioned above. Causal Shapley values (Heskes et al., 2020) seem to satisfy the mentioned desiderata. What are the benefits of SCE over Causal Shapley values? Are there any trade-offs?

**Questions:**

- Are there ways to satisfactorily resolve the counterexamples to SCE that I provided above?
- Please see other questions in the weakness section.